# How to Get Your LLM to Generate Challenging Problems for Evaluation

## Abstract

The pace of evolution of Large Language Models (LLMs) necessitates new approaches for rigorous and comprehensive evaluation. Traditional human annotation is increasingly impractical due to the complexities and costs involved in generating high-quality, challenging problems, particularly for tasks such as long-context reasoning. Moreover, the rapid saturation of existing human-curated benchmarks by LLMs further necessitates the need to develop scalable and automatically renewable evaluation methodologies. In this work, we introduce **CHASE**, a unified framework to synthetically generate challenging problems using LLMs without human involvement. For a given task, our approach builds a hard problem in a bottom-up manner from simpler components. Moreover since we want to generate synthetic data for evaluation, our framework decomposes the generation process into independently verifiable sub-tasks, thereby ensuring a high level of quality and correctness. We implement CHASE to create evaluation benchmarks across three diverse domains: document-based question answering, repository-level code completion, and math reasoning. The performance of state-of-the-art LLMs on these synthetic benchmarks lies in the range of 40-60% accuracy, thereby demonstrating the effectiveness of our framework at generating hard problems. Our experiments further reveal that the Gemini models significantly outperform other LLMs at long-context reasoning, and that the performance of all LLMs drastically drops by as much as 70% when we scale up the context size to 50k tokens.

## 1 Introduction

In the past few years, we have witnessed the emergence of powerful Large Language Models (LLMs) (OpenAI Team et al., 2024; Llama Team et al., 2024; Gemini Team et al., 2024) that exhibit remarkable performance over a wide range of tasks. However, the methods and resources for evaluating these models have not kept pace with their rapid evolution and increased capabilities. Contemporary LLMs have saturated many existing reasoning benchmarks (Chen et al., 2021; Cobbe et al., 2021). Developing challenging problems for reasoning tasks, such as math reasoning or code generation, can be both expensive and time-consuming, especially for human annotators with average expertise. Moreover, there are some settings, such as tasks requiring long-context reasoning over hundreds of thousands of tokens, where the generation of high quality data by humans is impractical. Additionally, we are facing a significant depletion of interesting data that is publicly available (such as SAT exams, Olympiad problems, etc.) for creating benchmarks. Hence, we believe that the conventional approach of developing evaluation benchmarks through human annotation and existing Web content is limited, and it is worthwhile to explore using LLMs for generating evaluation data.

Synthetic data generation has emerged as a powerful paradigm in recent years driven by the widespread availability of cheaper and faster LLMs that can effectively follow instructions. The focus of most prior works, however, has been on improving models by generating synthetic data for pre-training (Gunasekar et al., 2023), instruction fine-tuning (Wang et al., 2023; Xu et al., 2024), alignment (Bai et al., 2022), or task-specific fine-tuning (Yue et al., 2024; Wei et al., 2024). In contrast, using synthetic data for evaluation and benchmarking has been relatively underexplored.

There are considerable advantages in using synthetic data for evaluation: it is comparatively inexpensive, highly scalable, and can be renewed periodically to mitigate contamination concerns. However, there are two main challenges: first, *how can we create **hard** and **realistic** problems?* and second, *how*

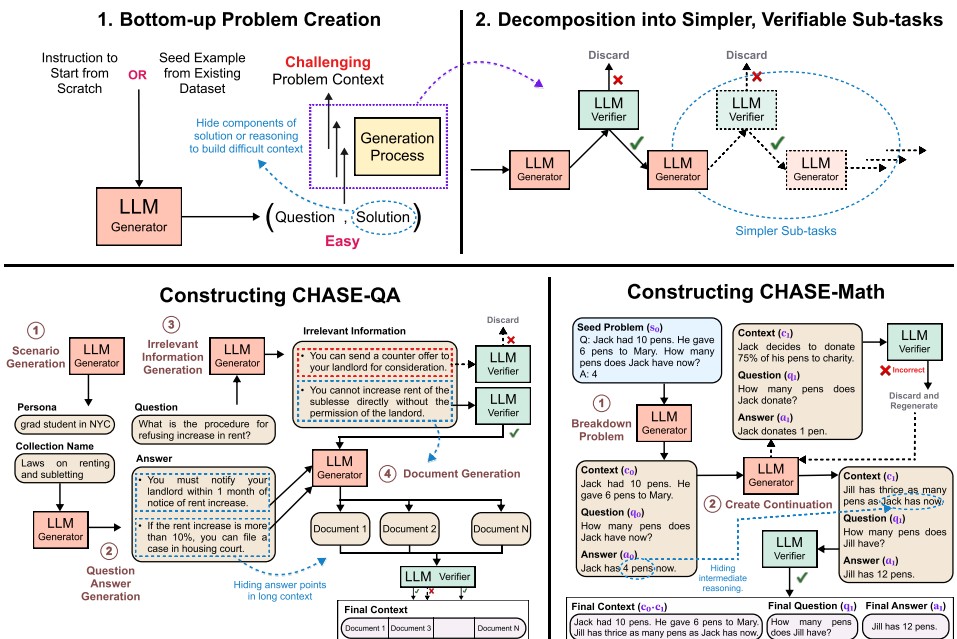

Figure 1: *Top:* Illustrating the high-level ideas behind our proposed CHASE framework. *Bottom left:* Pipeline for creating an example in CHASE-QA. *Bottom right:* Pipeline for creating a math word problem in CHASE-MATH. The pipeline for CHASE-CODE is illustrated in Figure 4 in Appendix.

*can we automatically **verify the correctness** of the generated data?* Typical synthetic data generation pipelines are either focused on self-improvement or on distillation. In the former, a model generates large-scale synthetic data that provides useful guidance for training. However, a large portion of such model-generated data is incorrect (Wang et al., 2023), which makes it difficult to adapt such pipelines for evaluation. Distillation-based approaches use stronger models to generate complex problems for training weaker models (Xu et al., 2024). However, we would like to design an approach that allows us to craft problems that are *challenging* to solve even for the LLM that generated them.

In this work, we present the **CHASE** framework: **CH**allenging **AI** with **S**ynthetic **E**valuations. Our methodology is based on two main ideas (see Figure 1) geared towards addressing the above-mentioned challenges. First, we create problems in a *bottom-up* manner where we iteratively hide parts of the solution within the problem's context. This makes problems challenging because finding the solution will require multiple steps of drawing inferences or reasoning over a longer context. Second, we decompose the generation process into simpler, *individually verifiable sub-tasks*. This facilitates fine-grained verification for correctness at each stage of the generation process.

We implemented our framework to create challenging benchmarks across three diverse domains. (1) **CHASE-QA** is a document-based question answering benchmark. The average length of documents for each problem is 6k tokens. This benchmark simulates the real-world application of answering user queries based on information that is spread across multiple documents, most of which may be irrelevant. (2) **CHASE-CODE** is a repository-level code completion benchmark with an average context length of 17k tokens. It consists of problems from two different domains: *data pre-processing* functions, and *algorithms*. This dataset simulates the real-world application of generating code in existing repositories based on user specifications. (3) **CHASE-MATH** is a grade-school level math word problems benchmark consisting of problems involving arithmetic reasoning. We built this benchmark to show the utility of our framework in building challenging problems for seemingly easy tasks on which models have already saturated existing benchmarks (Cobbe et al., 2021).

Experiments with **15** contemporary LLMs show that the datasets generated using CHASE are challenging for all models. The best performing LLMs only achieve accuracies in the range of $\sim \mathbf{40-60\%}$ across the three different domains. We further highlight the utility of CHASE by comparing it with direct prompting-based data generation baselines such as *Evol-Instruct* (Xu et al., 2024), which yield data with significant errors, apart from being relatively easier for models to solve.

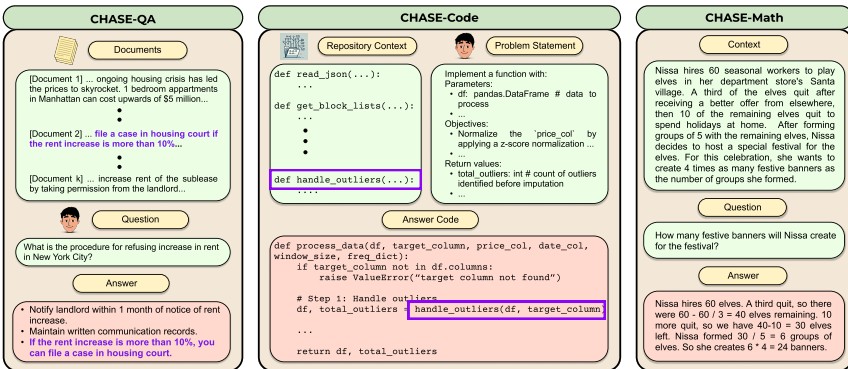

Figure 2: Examples of problems from all three benchmarks created using CHASE.

Our results also reveal large gaps in performance between different LLMs, all of which perform similarly on existing benchmarks like MMLU (Hendrycks et al., 2021) or HumanEval (Chen et al., 2021). Lastly, using our long-context benchmarks, we show that the performance of all LLMs decreases drastically (sometimes by upto 70%) when we increase the context size beyond 50k tokens.

## 2 RELATED WORK

**Synthetic data generation.** Prior works have explored generating synthetic data for various stages of an LLM's development pipeline: pre-training (Gunasekar et al., 2023; Ben Allal et al., 2024), instruction-finetuning (Wang et al., 2023; Xu et al., 2024), alignment (Bai et al., 2022; Lee et al., 2024) and task-specific fine-tuning (Wei et al., 2024; Yu et al., 2024a). The main focus of our work, however, is to generate high-quality challenging problems for evaluation and benchmarking. There is very limited existing literature in this area. Sprague et al. (2024) created a narrative-based question answering (QA) benchmark using a neuro-symbolic pipeline that first samples facts, and then uses an LLM to build a reasoning tree over them, along with generating a narrative. Bohnet et al. (2024) created another synthetic QA benchmark by extracting entities and their reference chains from existing stories and then prompting an LLM to generate questions over them. Gu et al. (2024) create a code understanding benchmark by prompting CodeLlama (Rozière et al., 2024) to generate python functions and their inputs and designing tasks based on predicting either the input or the output for a given function. In contrast to these works, we focus on presenting a general framework which can be used to design scalable pipelines to create challenging benchmarks across multiple domains. Moreover, we focus on realistic tasks such as information-seeking QA and repository-level code generation, both with extremely long contexts that we generate completely from scratch.

**Task-specific synthetic data.** Recent works have explored generating synthetic datasets for content-grounded QA tasks. Dai et al. (2022) use an LLM to develop information-seeking dialogue datasets based on text extracted from Wikipedia and the Web. Yehudai et al. (2024) prompt an LLM with wikipedia text to generate a question-answer pair. In contrast, we take a more application-oriented view and design a benchmark for document-based information-seeking questions that model realistic situations. Moreover, our pipeline generates the entire context (i.e., the documents) from LLMs, which allows a higher degree of control. There has also been significant interest in generating synthetic data for code. Yu et al. (2024b) employ an LLM-based framework to create examples for tasks based on existing raw code data. Wei et al. (2024) generate code instruction data by prompting an LLM with seed code snippets from existing repositories. Luo et al. (2024) generate complex and diverse code instruction-tuning data by applying the *Evol-Instruct* method (Xu et al., 2024) on existing code benchmarks. In this work, we focus on repository-level code completion, where we generate the repository contexts completely from scratch. Moreover, we also synthetically generate the corresponding test code to evaluate each example. Generating synthetic data to improve math reasoning has recently been a very active area of research. Previous work has explored generating new math problems by prompting LLMs with examples from existing datasets (Liu et al., 2023; Yu et al., 2024a; Lu et al., 2024). Similar to our work, some prior works have focused on creating challenging math problems. Shah et al. (2024) employ a human-in-the-loop approach to generate

novel and difficult math problems by prompting LLMs with multiple core skills that must be used in the problem. Liu et al. (2024a) employ iterative question composition where they iteratively prompt an LLM with a seed question to generate more complex variations of it. In this work, we design a completely automated pipeline to craft grade-school level math word problems that are challenging to solve even for the LLM that generated them.

## 3 THE CHASE FRAMEWORK AND BENCHMARKS

Our framework for generating synthetic data is based on two key ideas as illustrated in Figure 1.

**1. Bottom-up problem creation.**

We abandon the forward-thinking approach of first creating a difficult problem and then obtaining the corresponding solution, as followed by most works that create synthetic data for training (Xu et al., 2024; Liu et al., 2024a). If we first synthesize a complex problem and then obtain its corresponding solution from the generating LLM itself, then that problem is inherently solvable by that LLM. However, we wish to craft problems that are challenging even for the model which generates them. Hence, we instead take a different approach where we either generate or start with a simpler problem-solution pair, and then *bottom-up* build a challenging context. We make the problem's context challenging by systematically *hiding* components of the solution or reasoning such that they need to be either extracted from a long context or inferred based on given information.

**2. Decomposition into simpler, verifiable sub-tasks.**

We design pipelines that break down the generation process into simpler sub-tasks. Each individual LLM in the pipeline (i.e., each inference call) performs a simpler, specific function in the generation process. This provides us with multiple benefits. First, it grants us more control over each step of the generation process. We can treat each step as a task by itself and optimize the corresponding inference parameters individually. This also allows us to better manage the complexity and diversity of the generated data depending on our requirements. Second, and perhaps more importantly, it facilitates fine-grained verification. We deploy LLMs that are not part of the generation process to check the correctness and quality of the generated data at each possible step. We believe that LLMs can be relied upon for verification because our framework makes each verification task smaller and simpler compared to the main task of generating or solving the problem we are crafting.

We show the effectiveness and ease of adaptation of our framework by implementing it to create challenging problems across three diverse domains:

**CHASE-QA** is an open-domain document-grounded question answering task consisting of 671 problems. Each example in CHASE-QA consists of a set of documents and a question-answer pair, as illustrated in Figure 2 left. Models need to reason over a long context (more than 6k tokens) because the information relevant for answering the question is spread across multiple documents.

**CHASE-CODE** is a repository-level code completion benchmark consisting of 220 challenging problems. Given a repository of Python functions, the task is to implement a new function based on a set of objectives provided in natural language, as illustrated in Figure 2 centre. We create data for two broad domains: (1) *data pre-processing* operations such as dataframe manipulation, string processing, file I/O, etc., and (2) *algorithms* such as graph operations, array manipulations, etc.

**CHASE-MATH** is a challenging benchmark of 150 grade-school level math word problems involving only basic arithmetic operations. An example of the task is provided in Figure 2 right.

## 4 CONSTRUCTION PIPELINES

In this section, we discuss our implementation of the CHASE framework for all three domains in more detail. Our pipelines use two different LLMs: the generator **G**, and the verifier **V**.

### 4.1 CONSTRUCTING CHASE-QA

We generate CHASE-QA completely from scratch without relying on existing contexts or any seed examples from previous datasets. Following the CHASE framework, we create each example in a

bottom-up manner by first generating the question-answer pair, and then generating the corresponding documents. Our pipeline for creating CHASE-QA is illustrated in Figure 1 bottom left. We describe it in detail below. The exact prompts are provided in Appendix F.1.

**Generating diverse scenarios.** We begin by first generating a set of diverse realistic scenarios in which a *user persona* seeks to find some information from a *collection of documents*. An example would be a 'grad student in NYC' searching the 'laws on renting and subletting'. We prompt **G** to generate diverse scenarios in the form of a tuple (`persona`, `collection_name`) by bootstrapping it with 5 annotated example scenarios, and later prompting it with its own generated scenarios.

**Generating question-answer pairs.** Each scenario (`persona`, `collection_name`) defines the working environment for creating a particular example. We design programmatic prompts with a given scenario as the variable to prompt **G** to generate a realistic information-seeking question that the `persona` might want to know about from `collection_name` set of documents. For example, a 'grad student' might pose the question, 'what is the procedure for refusing increase in rent?', whose answer can be found spread across multiple documents about the government's laws on renting. Apart from the question, **G** must also generate the corresponding answer. We prompt **G** to generate questions and answers where the answers are a composition of multiple points or ideas. Additionally, **G** must generate the outline of the documents (only title and abstract) which will contain the answer. The idea is that it must separate out the answer points and assign them to these different documents.

**Generating irrelevant information.** To make the task more challenging, for each question-answer pair, we prompt **G** to generate other question-answer pairs where the answer is of a similar type or domain as the ground-truth answer. An example of a similar question for our running example with the grad student is 'what is the process to increase the rent for an appartment I am subletting?'. The intuition is that the corresponding answers to such similar questions will be of a very similar flavour to the ground-truth answer, but ultimately *irrelevant* for answering the question. This will make the generated data challenging since it will confuse the model when all of this similar type of information is spread across a long context. It is, however, important to verify that none of this newly generated irrelevant information is actually relevant for the original question (otherwise it will make our ground-truth answer incomplete). We individually prompt **V** with the original question and each of the supposed irrelevant information points to check if any part of them is relevant for answering the question (see Figure 1 bottom left for an example of an irrelevant point discarded by **V** because it was relevant for the original question).

**Generating documents.** For each example, we have generated a question-answer (QA) pair, along with some similar but irrelevant QA pairs. For each of these QA pairs, we separately prompt **G** to generate long documents where the documents must discuss the corresponding answer points assigned to it, along with many other irrelevant points. All documents discuss distinct information. Together, all these documents form up the context for that example. We verify two main things to ensure the correctness of the task: (1) none of the documents should contain any information relevant for answering the question, apart from the ground-truth answer points, and (2) all of the ground-truth answer points must be discussed somewhere in the documents. We do this by rigorously prompting **V** with individual documents and the ground-truth answer points.

## 4.2 Constructing CHASE-Code

We generate CHASE-Code completely from scratch without relying on existing contexts or any seed examples from previous datasets. Our pipeline for creating CHASE-Code is shown in Figure 4 in the Appendix. We describe it in detail below. The exact prompts are provided in Appendix F.2.

**Generating Python functions.** We begin by first generating a set of diverse and realistic Python functions. We prompt **G** to generate Python functions for a particular domain by bootstrapping it with 3 annotated functions in that domain, and later prompting it with its own generated functions. These generated functions will act as the *helper* functions in the repository context which may or may not be called in the answer code function. Given each generated helper function, we prompt **V** to generate a Python code which initializes sample inputs for the function and then calls it using them. We then execute this code to verify whether the generated helper function executes correctly.

**Generating problem statement and answer code.** To create a single example, we randomly sample $n$ of the previously generated helper functions, and prompt **G** to create a complex function that calls at least $k$ of these provided helper functions (hereafter called *relevant* helper functions) apart from implementing some other pieces of logic. This complex function is our *answer code*. Additionally **G** must elaborate in natural language what objectives the complex function achieves, which makes up our *problem statement*. Similar to the case of helper functions, we prompt **V** to generate test code to check if the generated answer code executes correctly. Moreover, we need to verify whether the generated problem statement sufficiently specifies the answer code. We prompt **V** with the problem statement and corresponding *relevant* helper functions and check whether the output is semantically equivalent to the answer code (using the test code obtained in the next step).

**Generating test code.** To enable automatic execution-based testing, we prompt **G** with the generated answer function to implement the test code for it in Python. The test code must independently implement the logic of the answer code without access to the corresponding helper functions. It must then initialize the parameters of the answer function with some sample values, and compare its output with its own implementation. We execute the generated test code to check if the corresponding answer code passes. We discard all examples for which (1) the test code does not execute properly, or (2) the test code executes but the answer code fails the test.

**Building code repository.** For each example, we build a unique repository of Python files. The repository consists of the *relevant* helper functions spread across different files, along with $m$ randomly sampled irrelevant Python functions from our previously generated set. The core difficulty of this task arises from understanding the entire long context of code functions, and identifying which ones are relevant for the provided problem statement.

### 4.3 CONSTRUCTING CHASE-MATH

We sample math word problems (MWP) from existing datasets as seed examples to build our benchmark. Following the CHASE framework, we bottom-up build a more complex problem by iteratively increasing the reasoning depth of the problem. An illustration of the pipeline used for creating CHASE-MATH can be seen in Figure 1 bottom right. We describe the pipeline in more detail below. The exact prompts are provided in Appendix F.3.

**Breaking down seed MWP.** A seed MWP $s$ is characterised by the tuple $s = (p, a)$ where $p$ is the problem, and $a$ is the answer. We prompt **G** to break down $p$ into two parts: the context $c$, which provides all the information, and the question $q$, which asks about some unknown quantity.

**Create continuation of MWP.** We prompt **G** with an initial seed MWP $s_0 = (p_0, a_0)$ to build a new problem which is a continuation of the previous problem. More precisely, **G** should output a new problem $s_1 = (p_1, a_1)$, where the context of $p_1$, i.e., $c_1$ assumes $a_0$ as given information (without explicitly stating it). For example, in Figure 1 bottom right, the model assumes *Jack has 4 pens* as given information, and creates a new continuation context, *Jill has thrice as many pens as Jack has now*. The model also generates a new question $q_1$, *how many pens does Jill have?* whose answer $a_1 = 12$ is obtained by performing an arithmetic operation (here, *multiplication by 3*) over $a_0 = 4$.

**Combining seed MWP with its continuation.** By combining the original seed problem with its continuation, we get a new MWP $s = (p, a)$ with a higher reasoning depth, where the context $c$ of the combined problem $p$ is a concatenation of the contexts of the seed problem and the continuation $c = c_0 \cdot c_1$. The question for the combined problem will be the one generated by the model, i.e., $q_1$, and the corresponding answer $a = a_1$. We refer to Figure 1 bottom right for illustration.

**Iteratively increase reasoning depth.** We increase the reasoning depth of a given seed MWP by creating new continuations in an iterative manner. Each new continuation $s_i$ formed after the $i^{\text{th}}$ iteration becomes the seed problem for the $(i + 1)^{\text{th}}$ iteration. The final complex problem after $j$ successful iterations, i.e., with a reasoning depth of $j + 1$, is given by context $c = c_0 \cdot c_1 \ldots c_j$, question $q_j$, and answer $a = a_j$.

Since each individual new problem created by **G** has a low reasoning depth of the same difficulty as the problems in the seed datasets, we verify their correctness using a non-identical ensemble of

verifier models $\{\mathbf{V_1}, \mathbf{V_2}, \ldots, \mathbf{V_n}\}$, each of which has very high performance on the seed dataset. We prompt each $\mathbf{V_k}$ with the generated problem context $c_i$ and question $q_i$ and check whether the prediction is the same as the generated answer $a_i$. If this fails for any verifier, we discard $s_i$ and begin generating again with $s_{i-1}$ as the seed MWP (see Figure 1 bottom right).

# 5 EXPERIMENTS

## 5.1 IMPLEMENTATION DETAILS

**Generating CHASE-QA.** We use GPT-4o (OpenAI Team et al., 2024) as the generator $\mathbf{G}$, and GPT-4o-mini as the verifier $\mathbf{V}$. We first sampled 500 different scenarios. For each scenario, we generate 2 different question-answer (QA) pairs. For each of the resulting 1000 unique QA pairs, we obtain *irrelevant* information by generating 4 similar QA pairs. We then generate the corresponding documents containing the ground-truth answer as well as irrelevant information for each of the 1000 examples. To increase the complexity of the resulting benchmark, we carry out a form of rejection sampling. We evaluate GPT-4o-mini twice on the task, and randomly discard half of the problems on which it was correct both times. This yielded the final benchmark of 671 examples.

**Generating CHASE-CODE.** We use GPT-4o-mini (OpenAI Team et al., 2024) as the generator $\mathbf{G}$, and Gemini-1.5-Flash as the verifier $\mathbf{V}$. We made this choice because generating even a small amount of challenging code problems required a large number of iterations, since a lot of the model-generated code at various stages would fail to execute or be semantically incorrect. For each domain, we first sampled 500 different helper functions that execute without errors. Then we prompt the model with $n = 10$ random helper functions to generate a problem statement and corresponding answer code that calls at least $k = 4$ helper functions. We do this to create 1000 different examples for each domain. Next, we generate up to 10 test codes for each example and keep only those examples for which a generated test code successfully passed for the corresponding answer code. We also carry out the verification of correctness of problem statement as describe before. This way, we end up with 290 examples for the *algorithms* domain and 300 examples for the *data pre-processing* domain. We again use GPT-4o-mini for rejection samping and randomly discard around half of the problems on which it was correct. This way, we end up with a total of 500 examples in the benchmark, with 250 examples for each domain. For each example, we randomly sample $m = 100$ *irrelevant* helper functions and distribute them into 10 Python files to constitute the repository context.

**Generating CHASE-MATH.** We again use GPT-4o-mini (OpenAI Team et al., 2024) as the generator $\mathbf{G}$, and an ensemble of Gemini-1.5-Flash and Llama-3.1-70B as the verifier $\mathbf{V}$. In practice, we observed that many of the model generated problems would fail at various stages of verification, so it is faster and cheaper to query the smaller models. We start with 2.3k seed problems taken from the test sets of GSM8k (Cobbe et al., 2021) and SVAMP (Patel et al., 2021). We set the maximum and minimum reasoning depth at 8 and 4 respectively. For each problem, we iterate 15 times to generate a problem continuation. Note that many of these iterations fail to produce a correct continuation of the problem, in which case we discard that generation and retry from that point in the subsequent iteration. We carry out this process 3 times. In this manner, we generated around 1500 problems. We then carry out rejection sampling and roughly discarded 75% of the problems that GPT-4o-mini could solve. In the end, we end up with a total of 500 challenging MWPs.

**Task parameters.** For CHASE-QA and CHASE-CODE, we prompt models with the instruction for the task, along with the corresponding long-context and question. The prompt formats are provided in Figure 19 and 28 respectively in Appendix F. For CHASE-MATH, we prompt models with the standard 8-shot chain-of-thought (Wei et al., 2022b). The prompt format is provided in Figure 31 in Appendix F.3. We decode for a maximum of 1024 tokens with a temperature of 0.5.

**Evaluation.** The ground-truth answers for CHASE-QA are verbose text, organized in bullet points. While this simulates real-world complexity, it also makes evaluation difficult. Since it is intractable to employ expert humans for evaluation, we deploy an LLM-as-a-judge to automatically assess the correctness of predictions. A prediction is considered to be correct if and only if it is (1) *complete*, i.e., it includes all the points mentioned in the ground-truth answer, and (2) *relevant*, i.e., it provides information only pertaining to the current question. We use GPT-4o as the judge and measure the

Table 1: The performance of various LLMs on all 3 domains of the CHASE benchmark. We measure the accuracy of the predictions for CHASE-QA and CHASE-MATH, and pass@1 for CHASE-CODE. DATA and ALGO refer to the *data pre-processing* and *algorithms* subdomains of CHASE-CODE. Numbers in **bold** indicate best performance on domain while underline indicates best-in-class performance.

| MODELS | QA | CODE | | MATH |
| | | DATA | ALGO | |
|---|---|---|---|---|
| Gemini-1.5-Pro | **63.2** | **35.6** | **40.8** | **65.4** |
| GPT-4o | 55.3 | 26.8 | 22.4 | 59.8 |
| Claude-3.5-Sonnet | 36.1 | 19.6 | 25.2 | 64.2 |
| Gemini-1.5-Flash | 55.1 | 25.6 | 31.6 | 56.6 |
| GPT-4o-mini | 50.2 | 19.6 | 18 | 48.4 |
| Claude-3-Haiku | 32.6 | 18 | 25.6 | 44.2 |
| Llama-3.1-70B | 41.3 | 12.4 | 18.8 | 53.4 |
| Mistral Large 2 | 34.1 | 4.8 | 5.2 | 59.6 |
| Qwen2.5-72B | 38.3 | 14.4 | 8.4 | 58.4 |
| Command R+ | 41.7 | 0 | 0 | 43.2 |
| DBRX | 15.7 | 1.2 | 3.2 | 21.6 |
| Phi-3.5-MoE | 10.6 | 0.4 | 1.2 | 39.4 |
| Mistral Small | 35.5 | 1.2 | 1.6 | 50.6 |
| Llama-3.1-8B | 25.2 | 0.8 | 3.2 | 32.2 |
| Qwen2.5-7B | 22.2 | 0 | 4.4 | 42.8 |

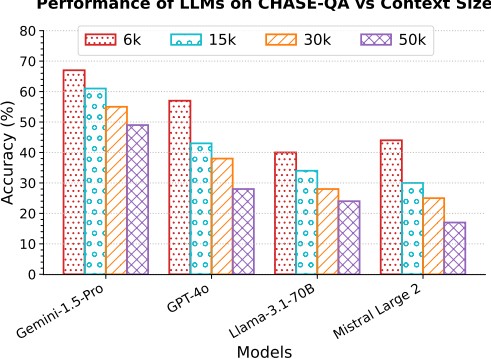

Performance of LLMs on CHASE-QA vs Context Size

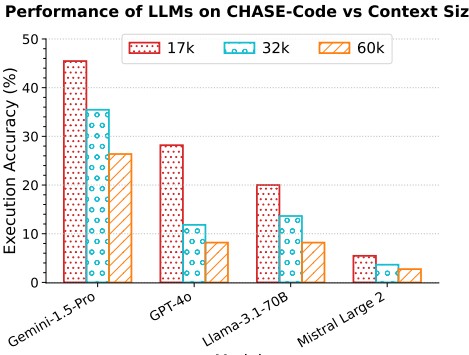

Performance of LLMs on CHASE-Code vs Context Size

Figure 3: Performance of LLMs decreases uniformly with increasing context sizes for the 100 example subset of CHASE-QA (*top*) and the 55 example subset of CHASE-CODE (*bottom*).

accuracy as the percentage of predictions judged to be correct. The prompt format used for evaluation is provided in Figure 20 in Appendix F.1. For CHASE-CODE, we measure the pass@1 execution accuracy, i.e., whether the model generated code correctly passes when we execute the corresponding test code in the first attempt. For CHASE-MATH, we measure the exact match accuracy against the ground-truth numerical answer.

**Models.** We evaluated a total of 15 different LLMs: Gemini-1.5-Pro and Flash (Gemini Team et al., 2024), GPT-4o and GPT-4o-mini (OpenAI Team et al., 2024), Claude-3.5-Sonnet Anthropic (2024b), Claude-3-Haiku (Anthropic, 2024a), Llama-3.1 8B and 70B (Llama Team et al., 2024), Mistral Small and Large 2 (Mistral, 2024), Qwen2.5 7B and 72B (Yang et al., 2024a; Team, 2024a), Cohere Command R+ (Cohere, 2024), DBRX-Instruct (Team, 2024b), and Phi-3.5-MoE (Abdin et al., 2024). The details of implementation are provided in Appendix B.

## 5.2 RESULTS AND DISCUSSION

**Performance of models.** Table 1 shows the performance of all 15 LLMs on all three bechmarks. For, CHASE-QA, all models, including the generator (and judge) GPT-4o, find the task challenging. The best performing model is Gemini-1.5-Pro which achieves only about 63% accuracy, suggesting massive room for improvement. All the models struggle even more on CHASE-CODE, with the best performing model only achieving 38.2% average accuracy. For CHASE-MATH, we see that even the most powerful LLMs only achieve 65.4% accuracy on a benchmark composed of grade-school level math word problems. Overall, these results clearly indicate the utility of the CHASE framework in crafting challenging problems that even state-of-the-art LLMs struggle to solve. We provide examples of errors made by Gemini-1.5-Pro on all three benchmarks and analyze them in Appendix E.

Table 2: Performance of LLMs on data generated by direct prompting approaches without using CHASE.

| MODEL | QA | MATH |
|---|---|---|
| Gemini-1.5-Pro | 81 | 85.7 |
| GPT-4o | 78 | 88.9 |
| Claude-3.5-Sonnet | 73 | 82.5 |

Table 3: Accuracy of LLMs increases marginally on CHASE-MATH when fine-tuned on data generated by Llama-3.1-8B.

| MODEL | BASE | FINE-TUNED |
|---|---|---|
| Llama-3.1-8B | 30 | 34.7 |
| Mistral-7B | 3.3 | 4.7 |
| Qwen2-7B | 12.7 | 15.3 |

On all three benchmarks, we see huge variations in performance between the models. Moreover, we observe models different from the generator and verifier (such as Gemini-1.5-Pro) that do better for all three tasks. This stands as a testament to the underlying quality of the generated data and highlights our framework's potential for differentiating between state-of-the-art LLMs that all perform similarly on standard benchmarks like MMLU (Hendrycks et al., 2021) or HumanEval (Chen et al., 2021). We further note some interesting observations. On both the long-context benchmarks, there is a substantial gap between the Gemini models and the other LLMs, clearly exhibiting the strong long-context reasoning capabilities of Gemini. Another interesting observation is that most of the models are stronger on the *algorithms* domain, while GPT-4o is stronger at *data pre-processing*. This demonstrates the utility of our benchmark at identifying such targeted differences in performance which could be very helpful for real tasks. For math reasoning, we have seen even weaker models like Llama-3.1-8B and Phi-3.5 get around $85-90\%$ accuracies on the popular GSM8k and SVAMP benchmarks. However, we see a large difference ($\sim 25-30\%$) between their performance and that of the state-of-the-art models, which may be attributed in part to some level of contamination of existing benchmarks (Zhang et al., 2024a).

**Direct generation baseline.** We experimented with directly prompting models to generate challenging data for the QA and math tasks, without using the CHASE framework. For QA, we prompt GPT-4o with unique examples from CHASE-QA as the seed task and instruct it to generate new examples in a manner similar to Honovich et al. (2023) and Wang et al. (2023). For math, we adapt the *Evol-Instruct* method (Xu et al., 2024) to generate more complex problems given seed examples from GSM8k. We carry out the same proportion of rejection sampling as we did for CHASE-QA and CHASE-MATH for fair comparison. We generated a total of 100 examples for both tasks. For the math task, we manually examined the generated problems and found that 34 of them had some kind of error such as the problem text being ambiguous or vague or the reasoning and answer being incorrect. Carrying out a detailed manual verification for the QA problems is impracticable, however, we believe it is highly likely that a significant portion of it is incorrect. We evaluated GPT-4o, Gemini-1.5-Pro, and Claude-3.5-Sonnet on these datasets and provide the results in Table 2. For both tasks, we observe that we are unable to generate challenging data by direct prompting.

**Impact of context size.** We studied the impact of varying the context size for long-context reasoning. For each example in a randomly-sampled 100-example subset of CHASE-QA, we increase the context size by concatenating the documents in that example with irrelevant documents randomly sampled from other examples. For CHASE-CODE, we create a subset of 55 randomly-sampled examples for each of the domains and increase the context size by concatenating irrelevant code functions in the corresponding repository context. Figure 3 plots the performances of 4 LLMs across different context sizes. For both benchmarks, we see a consistent and significant decrease in model performance as we scale up the context size. Hence, even though most modern LLMs have large context sizes (upwards of 128k), they still struggle to reason even at the scale of 30-40k tokens.

**Human verification of LLM judgements.** We measure the correlation of the GPT-4o evaluator's judgement and 3 human annotators over 100 randomly sampled[1] predictions made by Gemini-1.5-Pro on CHASE-QA. The accuracy of GPT-4o's judgement as measured against the majority vote of the annotators was 91%. Moreover, Cohen's kappa (Cohen, 1960) between the majority vote of the annotators and the LLM judge came out to be 0.82, which can be interpreted as almost-perfect agreement. Additional details regarding the setup of these experiments can be found in Appendix B.

---

[1]but balanced, according to GPT-4o's judgement.

**Fine-tuning smaller models.**   We study whether we can use much smaller models (of around 7B scale) to generate useful fine-tuning data for themselves following CHASE pipelines to perform better on evaluation benchmarks created by stronger models. We generate $\sim 10k$ math problems using Llama-3.1-8B as both the generator and the verifier and fine-tune 3 small models. Table 3 shows the accuracies on CHASE-MATH before and after fine-tuning on this dataset for all LLMs. We see marginal performance improvements across all models. These results seem to indicate that the evaluation data generated using significantly stronger models cannot be easily solved by such weak models even when fine-tuned on data generated by themselves using the exact same pipeline.

## 6   CONCLUSION

In this work, we presented CHASE, a framework to synthetically generate challenging problems for evaluation. Our framework offers multiple benefits. First, it is scalable and can efficiently generate hundreds of challenging problems. Second, it is renewable, which means it can be used to keep regenerating new datasets for a given task at periodic intervals thereby mitigating contamination concerns which plague most evaluation benchmarks. Third, it can effectively help evaluate tasks (such as long-context reasoning) which are inherently difficult or impracticable for humans to evaluate. Fourth, it provides a very high level of quality and correctness because of extensive verification. And finally, it can be used to generate *difficult* examples for a given task. We implemented CHASE on three different domains: document-based question answering, repository-level code completion, and math reasoning. With the help of extensive experiments, we showed that CHASE successfully generates examples that are difficult even for the state-of-the-art LLMs to solve. Our results raise several interesting questions to explore in future work: (a) How can we modify this framework to more easily adapt to different tasks? (b) What is the utility of this framework in generating data for training LLMs? (c) How can we verify LLM generations more efficiently? We hope our work will catalyze many such studies on using synthetic data for evaluation.

## 7   LIMITATIONS

**Size of benchmarks.**   The datasets we generated are comparatively smaller in size. Our framework necessitates querying the generator and especially the verifier many times for crafting each example. While this increases the quality and correctness of the data, it significantly increases the cost of generation. Moreover, a large portion of the intermediate generations in our pipeline are discarded because of extensive verification, which significantly reduces the yield. Our focus in this work is to present the CHASE framework and we believe our experiments, albeit on smaller-sized datasets, convincingly show its utility in generating challenging problems for evaluation. Small benchmarks, if generated with a high bar for quality and correctness, can still be very impactful as evidenced by past works like HumanEval (Chen et al., 2021).

**Instance-level issues.**   Some of the examples we generated using CHASE, while being semantically correct, use unnatural or difficult-to-parse language. This is a general trait of text generated from contemporary LLMs, and our framework is unfortunately susceptible to it. Moreover, while we carry out extensive verification of correctness at the instance-level using LLMs, it is still possible that some of the generated examples are incorrect. Verifying long-context reasoning data using non-expert humans is impracticable. To get a sense of the quality, we manually examined 30 random examples each from the generation process of CHASE-QA and CHASE-CODE. We found only 2 errors in CHASE-QA and none in CHASE-CODE. We also examined 100 examples from CHASE-MATH's pipeline and found 7 that were incorrect. Examples of such errors in generation are discussed in Appendix E. We believe that errors at this scale are acceptable considering the other advantages of the framework and can be factored into the evaluation in future studies.

**Adaptability.**   While we have shown how we implemented CHASE on three different domains, it is not trivial to adapt the framework to other tasks. Although the high level ideas behind CHASE are easy enough to follow, it takes multiple trials and errors to design a working pipeline for any given task. However, we are optimistic that advances in LLMs' abilities to more precisely follow instructions will make such pipelines easier to construct in the future.

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

## A ROADMAP

The appendix is organized as follows.

- In Section B, we provide the implementation details for our experiments.
- In Section C, we discuss some additional experimental results.
- In Section D, we discuss some additional related work.
- In Section E, we analyze errors made by LLMs while generating and solving CHASE benchmarks.
- In Section F, we provide the exact prompts used in this work.

## B IMPLEMENTATION DETAILS

Our code is implemented in PyTorch (Paszke et al., 2019) and makes use of the HuggingFace Transformers library (Wolf et al., 2020) and the vLLM library (Kwon et al., 2023) for running efficient inference locally on LLMs. All experiments with open models were done on our cluster with 8 NVIDIA A6000 GPUs with 48 GB memory. Experiments using GPT-4o and GPT-4o-mini were carried out using the OpenAI API.[2]. Experiments using Gemini-1.5-Pro and Gemini-1.5-Flash were carried out using the Google AI Studio.[3] Experiments with Claude-3.5-Sonnet and Claude-3-Haiku were carried out using Anthropic's API.[4] We provide the exact identifier and version for each LLM we experimented with in Table 4.

Table 4: Model identifiers for the 17 models we studied in our work. Models that are openly available are provided with links to their corresponding pages on Huggingface Hub.

| MODEL | EXACT IDENTIFIER |
|---|---|
| Llama-3.1-8B | meta-llama/Llama-3.1-8B-Instruct |
| Llama-3.1-70B | meta-llama/Llama-3.1-70B-Instruct |
| Mistral-7B | mistralai/Mistral-7B-Instruct-v0.1 |
| Mistral Small | mistralai/Mistral-Small-Instruct-2409 |
| Mistral Large 2 | mistralai/Mistral-Large-Instruct-2407 |
| Qwen2-7B | Qwen/Qwen2-7B-Instruct |
| Qwen2.5-7B | Qwen/Qwen2.5-7B-Instruct |
| Qwen2.5-72B | Qwen/Qwen2.5-72B-Instruct |
| Command R+ | CohereForAI/c4ai-command-r-plus-08-2024 |
| DBRX | databricks/dbrx-instruct |
| Phi-3.5-MoE | microsoft/Phi-3.5-MoE-instruct |
| GPT-4o-mini | gpt-4o-mini-2024-07-18 |
| GPT-4o | gpt-4o-2024-05-13 |
| Gemini-1.5-Flash | gemini-1.5-flash-001 |
| Gemini-1.5-Pro | gemini-1.5-pro-001 |
| Claude-3-Haiku | claude-3-haiku-20240307 |
| Claude-3.5-Sonnet | claude-3-5-sonnet-20240620 |

---

[2]https://platform.openai.com
[3]https://aistudio.google.com
[4]https://console.anthropic.com

Figure 4: Pipeline for creating an example in CHASE-CODE.

Table 5: Estimated cost of creating the benchmarks in terms of inference time and money.

| BENCHMARK | INFERENCE TIME (HOURS) | COST (USD) |
|---|---|---|
| CHASE-QA | 40 | 100 |
| CHASE-CODE | 55 | 150 |
| CHASE-MATH | 200 | 40 |

**Fine-tuning.** We use the torchtune library and fine-tune using LoRA (Hu et al., 2022). We did not extensively tune the hyperparameters as that is not the focus of this work. We used the following hyperparameters:

- Learning rate: 5e-4
- LoRA rank: 16
- LoRA alpha: 32
- Batch size: 2
- Scheduler: Cosine
- Precision: Brain `float16` (i.e., `bf16`)
- Optimizer: AdamW (Loshchilov & Hutter, 2019)

**Human verification of LLM judge.** We carry out human verification on Amazon Mechanical Turk. We first randomly sampled 10 of the predictions made by Gemini-1.5-Pro on CHASE-QA and manually evaluated them. We then publish them as a batch of 10 Human Intelligence Tasks (HITs) to serve as a qualification task to identify workers who will do the task properly. Note that each model prediction that needs to be judged is a HIT. Once we identified 3 workers that did perfectly on our qualification task, we published a batch of 100 randomly sampled predictions accessible only to those workers. Note that we sampled a balanced set based on the LLM judge's evaluation: 50 that were marked by GPT-4o as correct and 50 that were marked as incorrect. The instructions provided to the workers and the setup of the task is kept exactly the same as the one provided to the LLM judge as shown by the prompt in Figure 20. We paid $0.5 USD to the workers for every example.

**Cost of creation.** In Table 5, we report the estimated cost of creating the three benchmarks, both in terms of inference time and API expenses. Note that the inference time assumes sequential execution of each part of the pipeline with only one process running at a time. Hence, the generation can be made considerably faster with increased parallelism. This table does not include the cost of other experiments in the paper nor does it include the cost of background experiments that went into designing the pipelines. We estimate the total of these costs to be over $1000 USD.

Table 6: Comparison of model performances (pass@1) on CHASE-CODE and HumanEval, a widely-used benchmark for code generation.

| MODEL | CHASE-CODE | HUMANEVAL |
| --- | --- | --- |
| Gemini-1.5-Pro | 38.2 | 84.1 |
| GPT-4o | 24.6 | 90.2 |
| Claude-3.5-Sonnet | 22.4 | 92.0 |
| Gemini-1.5-Flash | 28.6 | 74.3 |
| GPT-4o-mini | 18.8 | 86.6 |
| Claude-3-Haiku | 21.8 | 75.9 |
| Llama-3.1-70B | 15.6 | 80.5 |
| Mistral Large 2 | 5.0 | 92.1 |
| Qwen2.5-72B | 11.4 | 86.6 |
| Command R+ | 0 | 70.1 |
| DBRX | 2.2 | 70.1 |
| Phi-3.5-MoE | 0.8 | 70.7 |
| Mistral Small | 1.4 | 73.8 |
| Llama-3.1-8B | 2.0 | 72.6 |
| Qwen2.5-7B | 2.2 | 57.9 |

# C ADDITIONAL RESULTS AND DISCUSSION

## C.1 COMPARISON OF MODEL PERFORMANCES ON SIMILAR DATASETS

**CHASE-QA** consists of long-context realistic-situation-based information-seeking QA problems. The most similar benchmarks are Loong (Wang et al., 2024b), which consists of long-context QA problems requiring reasoning over documents (more than 100k tokens long) from domains such as academic papers and financial reports, and LooGLE (Li et al., 2024b), which consists of long-dependency QA problems over wikipedia and movie scripts (around 32k tokens context). The best performing models on these datasets achieve scores of around 53% and 54% respectively. The best performing model on CHASE-QA achieves a score of around 63%, which reduces to around 55% when we scale the context size to comparable levels of 30k tokens.

**CHASE-CODE** consists of repository-level code generation problems. HumanEval (Chen et al., 2021) is the most widely-used challenging code generation benchmark. We compare the performances of all models on both datasets in Table 6. We can clearly see that CHASE-CODE is a much more challenging benchmark. Recently, some repository-level code benchmarks have also been proposed. SWE-Bench (Jimenez et al., 2024) is a benchmark of around 2300 software engineering problems compiled from GitHub issues in popular repositories. EvoCodeBench (Li et al., 2024a) consists of 275 repository-level code generation problems based on popular GitHub repositories. The best performing models on these benchmarks achieve around 42% and 20% scores respectively.

**CHASE-MATH** consists of grade-school level math word problems. The most widely-used challenging benchmark for this task is GSM8k (Cobbe et al., 2021), comprising of 1319 examples. We compare the performances of all models on both datasets in Table 7. It is clear that GSM8k has mostly become saturated, with many state-of-the-art models achieving more than 90% accuracies. In comparison, CHASE-MATH is still very difficult for all models to solve. Moreover, the differences in performance between different models is much larger, which enables more confident comparison.

Table 7: Comparison of model performances on CHASE-MATH and GSM8k, a widely-used benchmark for grade-school level math word problem solving.

| MODEL | CHASE-MATH | GSM8K |
|---|---|---|
| Gemini-1.5-Pro | **65.4** | 90.8 |
| GPT-4o | 59.8 | 96.1 |
| Claude-3.5-Sonnet | 64.2 | **96.4** |
| Gemini-1.5-Flash | 56.6 | 86.2 |
| GPT-4o-mini | 48.4 | 94.2 |
| Claude-3-Haiku | 44.2 | 79.2 |
| Llama-3.1-70B | 53.4 | 95.1 |
| Mistral Large 2 | 59.6 | 92.7 |
| Qwen2.5-72B | 58.4 | 95.8 |
| Command R+ | 43.2 | 70.7 |
| DBRX | 21.6 | 72.7 |
| Phi-3.5-MoE | 39.4 | 88.7 |
| Mistral Small | 50.6 | 87.4 |
| Llama-3.1-8B | 32.2 | 84.5 |
| Qwen2.5-7B | 42.8 | 85.4 |

## C.2 ALTERNATIVE METRICS OF EVALUATION FOR CHASE-QA

The metric of accuracy for CHASE-QA punishes models for not being concise and generating too many answer points that are not a part of the ground-truth answer. In this section, we present our experimental results with other softer evaluation metrics. We adapt two metrics that have been used by previous works for open-domain question answering (Adlakha et al., 2022): (1) **K-Precision**, which for a particular example, evaluates whether all of the answer points in the model's prediction are discussed in the documents, and (2) **Recall**, which evaluates whether all the ground truth answer points are a part of the model's prediction. K-Precision is used to measure the *faithfulness* of the model's prediction to the provided documents. Recall is used to measure the *correctness* of the model's prediction compared to the ground-truth. We define both the metrics as binary per example. Similar to how we calculated accuracy, we use GPT-4o as a judge with the prompts provided in Figure 21 and Figure 22 respectively. The results are provided in Table 8.

Note that the errors in CHASE-QA pertain to the cases where the ground-truth answer may not completely encompass all the relevant information about the question that is mentioned in the documents. We believe that comparisons of models on the basis of recall is relatively less affected by the presence of such errors. This is because if a model has comparatively lesser recall, that means that it generated more responses where it did not include the ground-truth information (irrespective of whether it generated any extra relevant information for the question that is not in the ground truth).

## C.3 EFFECT OF PROMPT FOR SOLVING CHASE-MATH

Considering the fact that CHASE-MATH is built by increasingly concatenating problems, we experiment with solving it using a different prompt format that explicitly instructs the model to process one sentence at a time, from the first to the last until it arrives at the final answer. We also illustrate this methodology in the prompt using 8 problems different from the original chain-of-thought prompt examples. Each of these new problems have a much higher reasoning depth. The prompt is provided in Figure 32. The results for 3 different models are shown in Table 9. While there is a clear increase in performance for all models, the task still remains difficult to solve, in general.

Table 8: Measuring performance of all models on CHASE-QA with alternative soft metrics, K-Precision and Recall.

| MODEL | ACCURACY | K-PRECISION | RECALL |
|---|---|---|---|
| Gemini-1.5-Pro | **63.2** | 85.1 | **68.6** |
| GPT-4o | 55.3 | **86.7** | 58.3 |
| Claude-3.5-Sonnet | 36.1 | 77.6 | 49.0 |
| Gemini-1.5-Flash | 55.1 | 82.3 | 61.7 |
| GPT-4o-mini | 50.2 | 74.1 | 50.7 |
| Claude-3-Haiku | 32.6 | 70.9 | 40.9 |
| Llama-3.1-70B | 41.3 | 76.3 | 46.1 |
| Mistral Large 2 | 34.1 | 72.4 | 42.9 |
| Qwen2.5-72B | 38.3 | 78.2 | 47.9 |
| Command R+ | 41.7 | 71.7 | 47.4 |
| DBRX | 15.7 | 53.2 | 35.0 |
| Phi-3.5-MoE | 10.6 | 45.0 | 25.6 |
| Mistral Small | 35.5 | 77.2 | 41.1 |
| Llama-3.1-8B | 25.2 | 61.3 | 32.0 |
| Qwen2.5-7B | 22.2 | 56.9 | 30.3 |

Table 9: Effect of prompt (see Figure 32) that explicitly instructs the model to solve CHASE-MATH problems by processing one sentence at a time.

| MODEL | 8-SHOT CoT (DEFAULT) | 8-SHOT SENTENCE-BY-SENTENCE |
|---|---|---|
| Gemini-1.5-Pro | 65.4 | 69.2 |
| GPT-4o | 59.8 | 61.4 |
| Llama-3.1-70B | 53.4 | 56.8 |

Examples of errors made by models even with this better prompting technique are provided in Figure 8 and Figure 9.

# D  ADDITIONAL RELATED WORK

## D.1  SYNTHETIC DATA GENERATION

Early works explored the potential of pretrained generative language models such as BERT (Devlin et al., 2019), GPT2 (Radford et al., 2019), and GPT-3 (Brown et al., 2020) to generate datasets for fine-tuning (Schick & Schütze, 2021; Ye et al., 2022) or for data augmentation (Kumar et al., 2020; Yoo et al., 2021). With increasingly powerful LLMs (Ouyang et al., 2022; OpenAI Team et al., 2024) being released in recent years combined with the benefits obtained from instruction fine-tuning (Wei et al., 2022a; Chung et al., 2024), the focus has shifted more towards generating synthetic instructions data. Honovich et al. (2023) and Wang et al. (2023) prompt GPT-3 with seed examples to automatically generate a large set of diverse tasks. Xu et al. (2024) introduced the *Evol-Instruct* pipeline to generate more complex examples from a given seed example. Mukherjee et al. (2023) leverage explanation traces from GPT-4 to create a large synthetic instruction-tuning dataset for

fine-tuning smaller LLMs. Mitra et al. (2024) design an agentic framework that uses raw text and code data as seeds to generate large scale synthetic data of prompts and responses.

Apart from synthetic instructions datasets, there has also been interest in using synthetic data for pre-training. Gunasekar et al. (2023) and Ben Allal et al. (2024) generate high-quality textbooks for pre-training small LLMs to unlock better reasoning performance. Recent works (Bai et al., 2022; Lee et al., 2024) have also explored using synthetic preference data for aligning language models. Synthetic data has started to become a major component in the post-training development phase of contemporary LLMs (Llama Team et al., 2024) to improve their reasoning capabilites. For a comprehensive discussion of major ideas and issues in synthetic data generation, we refer the reader to Liu et al. (2024b)'s survey.

### D.2 SYNTHETIC DATA FOR CODE GENERATION, AND MATH REASONING

There has been significant recent interest in generating synthetic data for code. Wen et al. (2024) utilize I/O specification apart from synthetic intents to generate data science code instructions. Patel et al. (2024) generate code library specifications using GPT-4 to evaluate whether LLMs can learn new code libraries in-context.

Generating synthetic data to improve math reasoning has also been a very active area of research. Tang et al. (2024) explored extracting topics and knowledge from math problems to prompt an LLM to generate new data. Toshniwal et al. (2024) create an instruction-tuning dataset by synthesizing code-interpreter style solutions for existing math problems using open-source LLMs. Yang et al. (2024b) create a reverse socratic approach to synthesize math problems from optimization scenarios. Luo et al. (2023) generate diverse math instructions data using *Evol-Instruct* and then train LLMs using reinforcement learning. Yue et al. (2024) build an instruction-tuning dataset by using GPT-4 to generate hybrid CoT and PoT rationales for examples from diverse math datasets. Yu et al. (2024a) bootstrap problems from existing math datasets by using an LLM to rephrase the question text.

### D.3 HUMAN-CURATED EVALUATION BENCHMARKS

**Question Answering.** There have been numerous context-grounded question answering benchmarks proposed in the past. These include SQuAD (Rajpurkar et al., 2016; 2018), HotpotQA (Yang et al., 2018), DROP (Dua et al., 2019), HAS-QA, Qasper (Dasigi et al., 2021), (Pang et al., 2019) TopioCQA (Adlakha et al., 2022), NovelQA Wang et al. (2024a), and RepliQA (Monteiro et al., 2024). Different from these works, we focus on extremely long contexts, with documents containing a total of more than 10,000 tokens. Another important aspect of our work is that all the documents that make up the context for an example are generated by an LLM from scratch, which reduces contamination risk. Moreover, our questions are designed to simulate everyday queries that users might ask an LLM-based agent. Correspondingly, the annotated ground-truth answers are at times very verbose, which make them impossible to objectively evaluate using automated metrics.

**Long Context Reasoning.** Kamradt (2023) introduced the Needle-In-a-Haystack (NIH) task in which a model is asked to retrieve a fact placed in a long context. However, this only assessed surface-level long-context retrieval abilities without much reasoning. In the past few months, many more long-context understanding benchmarks have been created. SCROLLS Shaham et al. (2022) and ZeroScrolls (Shaham et al., 2023) are long-context benchmarks created by aggregating and repurposing existing datasets. Compared to the QA datasets in ZeroScrolls which consist of contexts based on science and literature, CHASE-QA focuses on real-world scenarios where a user is looking for information across multiple documents. Many benchmarks such as LongBench (Bai et al., 2024), RULER (Hsieh et al., 2024), L-Eval (An et al., 2024), LooGLE (Li et al., 2024b) and InfiniteBench (Zhang et al., 2024b) consist of long-context QA tasks. However, the relevant information for answering the question is comparatively much more centralized in the context, surrounded by distracting information. In contrast, the relevant information in CHASE-QA is spread across multiple documents. Moreover, the irrelevant information in the other documents is still very closely related to the topics of the question, thereby making the task very difficult. Perhaps, the benchmark most similar to our work is Loong (Wang et al., 2024b). Like CHASE-QA, Loong also has evidences for answering questions scattered across a long context of documents. Unlike our benchmark, however, every document in Loong is known to be relevant for answering the question.

In contrast CHASE-QA models the more realistic scenario of searching a bunch of documents that are on very closely related topics, only some of which contain the answer. Moreover, unlike all these prior works, we create long-context reasoning benchmarks completely from scratch in which the long text contexts are also generated using LLMs. Further note that in contrast to these prior works that use manual annotation, our approach is highly scalable, capable of creating thousands of examples across diverse domains automatically in a much more cost-efficient manner.

# E   ERROR ANALYSIS

We provide examples for two types of errors, those made while solving the benchmarks, and those made while generating the benchmarks.

## E.1   ERRORS MADE WHILE SOLVING CHASE BENCHMARKS.

Figure 5 provides an example of an error made by Gemini-1.5-Pro on a problem from CHASE-QA. The model fails to mention two important points relevant for answering the question, which have been discussed in the documents. This provides a qualitative example of how even the most powerful models are unable to properly pay attention to all parts of a long-context and may miss some important information.

Figure 6 provides an example of an error made by Gemini-1.5-Pro in generating the correct code for a problem in CHASE-CODE. The model generates most of the code correctly, but for a particular objective, it gets confused in choosing to call the right helper function from the long-context code repository. This example qualitatively illustrates that doing well on this task requires not only a good understanding of the user-specified objectives, but also requires an in-depth understanding of all parts of the code repository.

Figure 7 provides an example of an error made by Gemini-1.5-Pro in solving a math word problem from CHASE-MATH. The model executes most of the reasoning steps correctly but fails at the last one. This example qualitatively shows how even state-of-the-art LLMs start to struggle when we scale up the reasoning depth of such types of problems.

## E.2   ERRORS MADE IN THE GENERATION PROCESS WHEN USING CHASE.

In Figure 10, we show an error made in the generation process of CHASE-QA by GPT-4o. In the document generation stage, the model generated a document which contained extra information that was directly relevant for answering the given question but was not included in the ground-truth answer. This is also a failure case of our verification engine (the one that uses the prompt in Figure 17) which failed to detect the presence of this extra relevant information in the generated document. We believe such errors can be further reduced by using an ensemble of verifiers to carry out each verification task.

Figure 11 provides en example of an error made by GPT-4o-mini while generating the CHASE-MATH benchmark. The model's generated answer did not correspond to its generated problem. This is also a failure for both the generator and verifier. Such failures may be reduced by using an ensemble of stronger LLMs for verification instead.

# F   PROMPTS

## F.1   PROMPTS FOR CHASE-QA

In this section, we outline the exact prompts for all experiments related to CHASE-QA:

- Generating scenarios: Figure 12
- Generating QA pairs: Figure 13
- Generating irrelevant information: Figure 14
- Generating documents: Figure 15
- Verifying irrelevant information: Figure 16

- Verifying absence of relevant information: Figure 17
- Verifying presence of answer: Figure 18
- Solving the task: Figure 19
- LLM-as-a-judge: Figure 20

## F.2 PROMPTS FOR CHASE-CODE

In this section, we outline the exact prompts for all experiments related to CHASE-CODE:

- Generating helper functions: Figure 23
- Generating problem statement and answer code: Figure 24
- Generating test code: Figure 25
- Generating test for verifying if function executes: Figure 26
- Verify correctness of problem statement: Figure 27
- Solving the task: Figure 28

## F.3 PROMPTS FOR CHASE-MATH

In this section, we outline the exact prompts for all experiments related to CHASE-MATH:

- Breaking down seed MWPs: Figure 29
- Creating continuation of MWP: Figure 30
- Solving MWP: Figure 31
- Solving MWP sentence-by-sentence: Figure 32

---

**Error Analysis: Solving CHASE-QA - Gemini-1.5-Pro**

**Question:** What are the regulations and financial support options available for starting an artisanal cheese-making business in Vermont?

---

**Ground-truth Answer:**
- Obtain a Dairy Processing License from the Vermont Agency of Agriculture, Food & Markets.
- Adhere to the Vermont Department of Environmental Conservation's (VT DEC) regulations regarding wastewater management.
- Apply for the Vermont Economic Development Authority (VEDA) Entrepreneurial Loan Program for startup financing.
- Comply with the Vermont Department of Taxes' requirements for sales and use tax on dairy products.
- Secure a Vermont Working Lands Enterprise Initiative Grant for equipment and facility improvement.

---

**Irrelevant Information:**
- Obtain a Microbrewery License from the New York State Liquor Authority (NYSLA).
- Adhere to the New York State Department of Environmental Conservation (NYSDEC) regulations regarding waste disposal.

[Redacted]

---

**Prediction:**
- Comply with the Vermont Department of Taxes' requirements for sales and use tax on dairy products, including registering for a sales tax permit, tracking sales, and remitting collected tax. (Document 10)
- Adhere to the Vermont Department of Environmental Conservation's (VT DEC) regulations regarding wastewater management, ensuring waste byproducts are treated and disposed of properly to prevent water contamination. (Document 13)
- Obtain a Dairy Processing License from the Vermont Agency of Agriculture, Food & Markets. (Document 12)

---

**LLM Evaluation:**
Result: False. The prediction is missing two main points from the ground-truth answer:
1. Applying for the Vermont Economic Development Authority (VEDA) Entrepreneurial Loan Program for startup financing.
2. Securing a Vermont Working Lands Enterprise Initiative Grant for equipment and facility improvement.

These are crucial financial support options that are not mentioned in the prediction.

Figure 5: Example of error made by Gemini-1.5-Pro on CHASE-QA. The GPT-4o judge correctly evaluates the prediction to be wrong because it fails to mention some major points from the ground-truth answer which were discussed in the documents.

**Error Analysis: Solving CHASE-CODE (Algorithms) - Gemini-1.5-Pro**

**Objectives:**

[Redacted]

- For each coordinate from the previous sub-goal, calculate the Manhattan distance from the origin (0,0) for the interesting cells found, returning a list of tuples.

**Ground-truth Answer Code:**

```python
import numpy as np
from matrix_decomposition import svd_reconstruction
from entity_extractor import extract_entities
from meeting_scheduling import can_schedule_meetings
from grid_analyzer import high_value_coordinates

def complex_analysis(matrix, k, text, entities, intervals,
        threshold):
    reconstructed_matrix = svd_reconstruction(matrix, k)
    extracted_entities = extract_entities(text, entities)
    can_schedule = can_schedule_meetings(intervals, 0)
    interesting_cells = high_value_coordinates(reconstructed_matrix
        , threshold)
    interesting_cells_with_distances = [((i, j), abs(i) + abs(j))
        for ((i, j), _) in interesting_cells]
    return reconstructed_matrix, extracted_entities, can_schedule,
        interesting_cells_with_distances
```

**Predicted Code:**

```python
import numpy as np
from matrix_decomposition import svd_reconstruction
from entity_extractor import extract_entities
from meeting_scheduling import can_schedule_meetings
from matrix_analyzer import find_interesting_cells

def complex_analysis(matrix, k, text, entities, intervals,
        threshold):
    reconstructed_matrix = svd_reconstruction(np.array(matrix), k)
    extracted_entities = extract_entities(text, entities)
    can_schedule = can_schedule_meetings(intervals, 0)
    interesting_cells = find_interesting_cells(reconstructed_matrix
        , threshold)
    interesting_cells_with_distances = [((row, col), abs(row) + abs
        (col)) for row, col in interesting_cells]
    return reconstructed_matrix, extracted_entities, can_schedule,
        interesting_cells_with_distances
```

Figure 6: Example of error made by Gemini-1.5-Pro on an example from the *algorithms* split of CHASE-CODE. The prediction incorrectly calls the find_interesting_cells helper function (see line number 11) which does not calculate manhattan distances as requested by the objective.

---

**Error Analysis: Solving CHASE-MATH - 8-shot chain-of-thought - Gemini-1.5-Pro**

**Question:**
Harry slept 9 hours last night. His friend James slept only 2/3 of what Harry slept. After their sleep, Harry decided to wake up early and went for a jog. He invited James to join him, and they jogged for 4 times the additional hours Harry slept compared to James. They both felt energized and decided to go for breakfast, where they calculated the total hours they spent jogging. After their breakfast, Harry and James sat down to review their fitness goals. They discussed their jogging session and decided it would be beneficial to repeat such sessions. However, they wanted to gauge how many smaller jogging sessions equal the time they previously spent jogging. James suggested that they break down their jogging time into segments of 3 hours to see how many complete segments they can fit within their total jogging time. After evaluating their jogging goals, Harry and James decided to increase the duration of their future jogging sessions. They considered adding an additional hour to each segment to further enhance their fitness regimen, while keeping the same total jog time of all sessions combined. After successfully completing their new jogging regimen for a month, Harry and James noticed that their stamina had increased significantly. They decided to celebrate this achievement by doubling their number of sessions for the future. They planned to increase from their current number of complete sessions to a total that is double their current sessions. They are excited to see how their increased devotion to jogging will impact their overall fitness. After a successful month of their increased jogging regimen, Harry and James wanted to keep track of their progress. They decided to add rest sessions into their training schedule to avoid burnout. For every 1/3 of the number of jogging sessions in their regimen, they would incorporate 1 rest session. If they have done 4 rest sessions so far, how many total jogging sessions have they finished?

---

**Ground-truth Answer:**
James slept 9 hours * 2/3 = 6 hours. So, Harry slept 9 - 6 = 3 hours more than James. They jogged for 4 times that amount, so 12 hours together. We then divide 12 by 3 to get 4 complete sessions. Now divide the 12 hours by the new session duration (4 hours) which yields 3 sessions. Doubling that quantity results 6. They add rest sessions for every 1/3 * 6 = 2 jogging sessions. So for 4 rest sessions, they have 4 * 2 = 8 jogging sessions. #### 8.0

---

**Prediction:**
Let's break down this problem step-by-step:
1. **James's sleep:** James slept 2/3 of Harry's 9 hours, which is (2/3) * 9 = 6 hours.
2. **Additional hours Harry slept:** Harry slept 9 hours - 6 hours = 3 more hours than James.
3. **Jogging time:** They jogged for 4 times the additional hours Harry slept, meaning they jogged for 4 * 3 = 12 hours.
4. **Segments of jogging:** They want to break down their jogging time into 3-hour segments, so they have 12 hours / 3 hours/segment = 4 segments.
5. **Increased segment duration:** They increase each segment by 1 hour, making each segment 3 hours + 1 hour = 4 hours long.
6. **Total sessions:** Since they keep the same total jogging time, they still have 12 hours of jogging. With the new segment length, they have 12 hours / 4 hours/session = 3 sessions.
7. **Doubled sessions:** They double their number of sessions from 3 to 3 * 2 = 6 sessions.
8. **Rest sessions and total jog sessions:** They incorporate 1 rest session for every 1/3 of their jogging sessions. Since they've done 4 rest sessions, they have completed 4 rest sessions * 3 jog sessions/rest session = 12 jogging sessions.
The answer is 12.

Figure 7: Example of error made in the last step by Gemini-1.5-Pro on an example from CHASE-MATH when prompted with the default 8-shot chain-of-thought.

---

**Error Analysis: Solving CHASE-MATH - 8-shot sentence-by-sentence - Gemini-1.5-Pro**

**Question:**
James decides to run 3 sprints 3 times a week. He runs 60 meters each sprint. After a month of consistent training, James feels he needs to increase his speed. He decides to take a break from sprinting for one week to focus on recovery. During this recovery week, he runs a shorter distance of 30 meters for only 2 sprints, 3 times that week instead. After his recovery week, James feels rejuvenated and ready to resume his training. He decides to continue running but at double the distance he covered during his recovery week for each day the next week, aiming to improve his overall performance. He runs this increased distance for 7 days straight, maintaining consistency every day. After running consistently throughout the week, James notices a small injury starting to develop in his leg. Concerned about this, he decides to cut back on his total running distance for the following week. To ensure he allows for proper recovery and does not worsen the injury, he plans to reduce his total running distance from the prior week by 420 meters. How much distance will James run in the following week after the reduction?

---

**Ground-truth Answer:**
He sprints $3 * 3 = << 3 * 3 = 9 >> 9$ times So he runs $9 * 60 = << 9 * 60 = 540 >> 540$ meters In the recovery week, James runs 2 sprints each day for 3 days, covering 30 meters each sprint. Therefore, the total distance covered during the recovery week is calculated as follows: he runs 2 sprints/day * 3 days = 6 sprints. Then, multiplying by the distance per sprint gives 6 sprints * 30 meters/sprint = 180 meters. To find out how much less he runs compared to his regular week, we subtract the total distance he runs that week from the distance he typically runs in a week, which is 540 - 180 = 360 meters less. James runs a distance that is double what he covered in the recovery week, which is 180 meters. Therefore, if he runs this doubled distance for 7 days, we first calculate the distance he covers in one day, which is 180 meters * 2 = 360 meters. Over the course of 7 days, the total distance will be 360 meters/day * 7 days. To find the distance James will run in the following week after reducing his initial total distance by 420 meters, I will need to subtract 420 from the total distance he ran in the previous week, which was 2520.0 meters. Hence, the calculation will be: 2520.0 - 420 = 2100.0 meters. ####2100.0

---

**Prediction:**
Sentence 1: James runs 3 sprints * 60 meters/sprint = 180 meters per day.
Sentence 2: He runs 180 meters/day * 3 days/week = 540 meters per week.
Sentence 3: Nothing to solve.
Sentence 4: During his recovery week, he runs 30 meters/sprint * 2 sprints = 60 meters per day.
Sentence 5: He runs this shorter distance 3 times a week, so 60 meters/day * 3 days/week = 180 meters.
Sentence 6: Nothing to solve.
Sentence 7: He doubles his recovery week distance, running 60 meters * 2 = 120 meters per day.
Sentence 8: He runs this distance for 7 days straight, totaling 120 meters/day * 7 days = 840 meters.
Sentence 9: Nothing to solve.
Sentence 10: He reduces his total running distance by 420 meters, meaning he'll run 840 meters - 420 meters = 420 meters.
The answer is 420.

Figure 8: Example of error made by Gemini-1.5-Pro on an example from CHASE-MATH when prompted with 8-shot sentence-by-sentence solving examples.

---

**Error Analysis: Solving CHASE-MATH - 8-shot sentence-by-sentence - Gemini-1.5-Pro**

**Question:**
Rani has ten more crabs than Monic, who has 4 fewer crabs than Bo. If Bo has 40 crabs, calculate the total number of crabs the three have together. After counting their crabs, Rani, Monic, and Bo decide to share their crabs equally among themselves for a crab feast. However, before they can do that, Rani finds out that they must provide 2 additional crabs to a local aquarium as a donation. After their crab feast, Rani, Monic, and Bo decide to host another gathering for their friends. They plan to invite 5 additional friends to enjoy crabs. Each person, including Rani, Monic, Bo, and their friends, should receive an equal amount of crabs for this new gathering. They go and buy new crabs which are in total 3 times the number of crabs each person ate after the donation. How many crabs did they have altogether to share with the additional friends in the end?

---

**Ground-truth Answer:**
If Bo has 40 crabs, then Monic, who has 4 fewer crabs than Bo, has 40-4 = $<< 40 - 4 = 36 >>$ 36 crabs. Bo and Monic have a total of 36+40 = $<< 36 + 40 = 76 >>$ 76 crabs. Since Monic has 36 crabs, Rani's number of crabs is 36+ 10 = $<< 36 + 10 = 46 >>$46. Together, the three have 76+46 = $<< 76 + 46 = 122 >>$ 122 crabs. First, calculate how many crabs remain after the donation: 122 - 2 = 120. Now, to find out how many crabs each person will receive, divide the remaining number of crabs (120) by 3, since there are three individuals: 120 3 = 40. Since each of them originally ate 40 crabs, if we multiply that number by 3, we calculate the total number of crabs available in the end. ####120.0

---

**Prediction:**
Sentence 1: Monic has 40 - 4 = 36 crabs.
Sentence 2: Rani has 36 + 10 = 46 crabs.
Sentence 3: Bo has 40 crabs.
Sentence 4: Together they have 46 + 36 + 40 = 122 crabs.
Sentence 5: They donate 2 crabs, leaving them with 122 - 2 = 120 crabs.
Sentence 6: They share the 120 crabs equally, so each person gets 120 / 3 = 40 crabs.
Sentence 7: They invite 5 more friends, making a total of 3 + 5 = 8 people.
Sentence 8: They buy new crabs totaling 3 times the number each person ate after the donation, which is 3 * 40 = 120 crabs.
Sentence 9: Including the 120 crabs they bought and the remaining crabs they shared after donating, they had 120 + 40*3 = 120 + 120 = 240 crabs altogether to share with the additional friends.

The answer is 240.

---

Figure 9: Example of error made by Gemini-1.5-Pro on an example from CHASE-MATH when prompted with 8-shot sentence-by-sentence solving examples. The model fails to understand that the 120 crabs they had earlier have already been eaten and that only the new 120 crabs bought in the end are available to share with the new friends.

1674
1675
1676
1677
1678
1679
1680
1681
1682
1683
1684
1685
1686
1687
1688
1689
1690
1691
1692
1693
1694
1695
1696
1697
1698
1699
1700
1701
1702
1703
1704
1705
1706
1707
1708
1709
1710
1711
1712
1713
1714
1715
1716
1717
1718
1719
1720
1721
1722
1723
1724
1725
1726
1727

---

**Error Analysis: Generating CHASE-QA - GPT-4o**

**Documents:**
[Redacted]

Document 2: Title: Semiconductor Advancements and Emerging Trends Text: The semiconductor industry has witnessed numerous advancements over the past few years, driven by the rapid pace of technological innovation. One of the most promising materials being explored for next-generation processors is Gallium Nitride (GaN). GaN is being investigated for its potential to significantly improve the speed and efficiency of processors. Its unique properties, such as high electron mobility and wide bandgap, enable faster electron transmission and greater thermal conductivity. This makes GaN an ideal candidate for applications requiring high power and high-frequency operation.

Developments in artificial intelligence algorithms for natural language processing continue to push the boundaries of what machines can understand and generate in human language. Meanwhile, the impact of 5G technology on mobile communications and IoT devices opens up new possibilities for high-speed, low-latency connectivity.

Additionally, **Graphene-based semiconductors hold the potential to revolutionize processor technology**. Graphene, a single layer of carbon atoms arranged in a hexagonal lattice, exhibits exceptional electrical properties. Its high electrical conductivity and mechanical strength make it an attractive material for creating faster and more efficient processors. Researchers are working on overcoming the challenges associated with mass production and integration of graphene into existing manufacturing processes.

[Redacted]

---

**Question:** What new semiconductor materials are being explored for next-generation processors, and which startups have recently secured funding to advance these technologies?

---

**Ground-truth Answer:**
- Gallium Nitride (GaN) is being investigated for its potential to improve the speed and efficiency of processors.
- Anokiwave, a pioneering startup in GaN-based semiconductors, has recently raised $45 million in Series C funding.
- Paragraf, a startup developing electronic devices for the semiconductor industry, secured $30 million in a recent funding round.
- The development of molybdenum disulfide (MoS2) as a semiconductor material could lead to significant advancements in processor miniaturization.

Figure 10: Example of an error made in the generation process of CHASE-QA. The model generated a document which contained additional relevant information about the question (the point about graphene-based semiconductors highlighted in bold in the text) which is not a part of the ground-truth answer.

---

**Error Analysis: Generating CHASE-QA - GPT-4o**

**Question:** Rani has ten more crabs than Monic, who has 4 fewer crabs than Bo. If Bo has 40 crabs, calculate the total number of crabs the three have together. After counting their crabs, Rani, Monic, and Bo decide to share their crabs equally among themselves for a crab feast. However, before they can do that, Rani finds out that they must provide 2 additional crabs to a local aquarium as a donation. After their crab feast, Rani, Monic, and Bo decide to host another gathering for their friends. They plan to invite 5 additional friends to enjoy the remaining crabs. Each person, including Rani, Monic, Bo, and their friends, should receive an equal amount of crabs for this new gathering. They want to figure out how many crabs they have available for their friends while ensuring that each person gets an equal share. If they multiply the number of crabs each had after the donation by 3, how many crabs did they have altogether to share with the additional friends?

---

**Ground-truth Answer:**
If Bo has 40 crabs, then Monic, who has 4 fewer crabs than Bo, has 40-4 = $<< 40 - 4 = 36 >>$ 36 crabs. Bo and Monic have a total of 36+40 = $<< 36 + 40 = 76 >>$ 76 crabs. Since Monic has 36 crabs, Rani's number of crabs is 36+ 10 = $<< 36 + 10 = 46 >>$ 46. Together, the three have 76+46 = $<< 76 + 46 = 122 >>$ 122 crabs. First, calculate how many crabs remain after the donation: 122 - 2 = 120. Now, to find out how many crabs each person will receive, divide the remaining number of crabs (120) **by 3, since there are three individuals**: 120 ÷ 3 = 40. Since each of them originally received 40 crabs, if we multiply that number by 3 individuals (Rani, Monic, and Bo), we calculate the total number of crabs available for sharing. ####120.0

Figure 11: Example of an error made in the generation process of CHASE-MATH. The reasoning in the answer wrongly considers the number of people for which to divide crabs to be 3 while the context has added new information that there are 5 additional friends to consider.

**CHASE-QA: Generating Scenarios**

**System Prompt:** You are an expert generator of data.

---

You are a research scientist. You want to make data to test an advanced question answering system.

Give me 5 examples of real-life scenarios where a USER_PERSONA may seek information in a COLLECTION_OF_DOCS. Do not consider educational or historical scenarios.

Some examples are:
USER_PERSONA: College student
COLLECTION_OF_DOCS: Intranet on the university website

USER_PERSONA: Intern doctor at a hospital
COLLECTION_OF_DOCS: Encyclopedia of diseases

USER_PERSONA: Immigrant in NYC
COLLECTION_OF_DOCS: Laws on renting and subletting

USER_PERSONA: HR manager at a top law firm
COLLECTION_OF_DOCS: Court and newspaper records

USER_PERSONA: Scientist at an NGO
COLLECTION_OF_DOCS: Government website for Income Tax

Answer in the following format:
USER_PERSONA:
COLLECTION_OF_DOCS:

Figure 12: Prompt for generating diverse scenarios for CHASE-QA.

**CHASE-QA: Generating QA Pairs**

**System Prompt:** You are an expert generator of data. Do not use ** to start lines or denote points.

---

You are a research scientist. You want to make data to test an advanced question answering system.

Give me an example question and corresponding answer that a {USER_PERSONA} may ask that compulsorily requires searching a {COLLECTION_OF_DOCS}. Make questions that cannot be answered directly with general knowledge but necessarily require some uncommon information that is present in some documents. The answer must be very specific and written in bullet points, so that it is easier to objectively evaluate. Depending on the question, the answer can have anything between 3-6 bullet points without any sub-points.

The answer to the question you create must be scattered across different documents (at least 3). Assign each point of the answer to a specific document in which that point will be discussed. You may assign multiple points to the same document, but each point must only be assigned to a single document. You must state the title and answer points assigned for each of the documents.

Answer in the following format:

Question: <Question>
Answer: <Answer>

Document 1 Title: <Title>
Document 1 Answer points assigned: <Points>

Document 2 Title: <Title>
Document 2 Answer points assigned: <Points>

and so on...

Figure 13: Programmatic prompt for generating question-answer pairs for CHASE-QA.

---

**CHASE-QA: Generating Irrelevant Information QA Pairs**

**System Prompt:** You are an expert generator of data. Do not use ** to start lines or denote points.

---

You are a research scientist. You want to make hard data to test an advanced question answering system. You are given a question that a {USER_PERSONA} might want answered, along with the corresponding answer, and information of documents from {COLLECTION_OF_DOCS} that are important for answering that question.

Original Question: {QUESTION}

Original Answer:
{ANSWER}

Original Documents Information:
{DOCS_INFORMATION}

You must generate an adversarial question, adversarial answer, and corresponding adversarial documents that ask for something different but on similar topics or type so that it is difficult to answer the original question. Examples of how adversarial questions should look like are provided below:

Original Question: What are the best activities to do in Montreal, Canada during the winter season?
Adversarial Question: What activities should I look at when visiting Tokyo during the summer?

[Redacted]

Also provide an answer to the adversarial question, which is similar in style to the original answer, but differs significantly in information or specifics. The answer points for the adversarial question should be written in context of that adversarial question, so that they cannot be confused with the original question. Note that none of the points appearing in the original answer should be present in the answer to the adversarial question.

The answer to the adversarial question you craft must be scattered across different documents (at least 3) separate from the original answer documents. Assign each point of the adversarial answer to a specific document in which that point will be discussed. You may assign multiple points to the same adversarial document, but each point must only be assigned to a single adversarial document. You must state the title and adversarial answer points assigned for each of the adversarial documents. These adversarial documents should not have any overlapping information with the original answer documents.

Answer in the following format:

[Redacted]

Figure 14: Programmatic prompt for generating irrelevant information question-answer pairs for CHASE-QA.

---

**CHASE-QA: Generating Documents**

**System Prompt:** You are an expert data generator. Following the instruction, you must generate long and correct documents.

---

You need to generate the documents for an example of a retrieval based Question Answering Task.

The task consists of n documents provided in English text that consist of information about different topics and a question. To answer the question correctly compulsorily requires using some of the information in some subset of the documents provided.

Given below is a situation faced by {USER_PERSONA} when searching {COLLECTION_OF_DOCS}. The question-answer pair is:

Question: {QUESTION}
Answer: {ANSWER}

Given below are the assigned answer points for each document.

{DOCS_INFORMATION}

Your job is to create long documents according to this information. For each document, first create 10-12 unique other points that are in no way related to the topic of the question and answer (different points for each document). These points should discuss very different things about a similar but different topic. Then use these points along with the assigned answer points to create a long document (at least 700 words long). The assigned answer points must be discussed taking into account the question. You must only discuss about these points and nothing else. Change the order of the points so that the answer points are embedded inside the document. Assign an appropriate title to the document. Do not summarize or conclude the document in the end.

Additionally, ensure that the documents you create do not have any information related to the following irrelevant question-answer pairs. You should create documents that discuss topics that are completely different from the following information.

{IRRELEVANT_QUESTIONS_ANSWERS}

Give output in the following format:
Document 1:
Title: <Title>
Question: {QUESTION}
Answer points assigned [Only these points must be covered with respect to the question]:
<Points>
Other unrelated points created: <Points>
Text:
<Document Text>

[Redacted]
and so on...

Figure 15: Programmatic prompt for generating documents for CHASE-QA.

---

**CHASE-QA: Verifying Irrelevant Information**

**System Prompt:** You are an expert at verifying data.

---

You are given a question and an answer. You must check whether the answer is even partially relevant for answering the question. If the answer is not relevant at all, output "False" to "Relevance". Otherwise, if and only if the answer discusses information that is at least partially necessary to answer the question, output "True".

Question: {QUESTION}

Answer:
{IRRELEVANT_ANSWERS}

Give output in the following format:
Relevance: True/False

---

Figure 16: Programmatic prompt for verifying irrelevance of irrelevant information for CHASE-QA.

---

**CHASE-QA: Verifying Absence of Relevant Information**

**System Prompt:** You are an expert at verifying data.

---

You are given a document followed by a question and some answer points. You must check whether there are any additional major points in the document that provide relevant information for answering the question that are currently missing from the answer. Follow these instructions:

1. Do not look for exact phrases or explicit mentions since the answer can have points that are a paraphrase of the same broad information.

2. It is ok if the document provides more specifics or details about the points already in the answer or if it discusses them in more depth by introducing related information so you can ignore that.

3. Check if the document introduces a new "major" idea or point that is crucial for answering the question and is not at all mentioned in the answer and is not an extension of the existing points in the answer.

4. Your job is not to check if the question can be sufficiently answered. You should ignore if the document or answer points are missing any points that are needed in the answer to the question.

If the document is not introducing major new points pertaining to the answer, output "False" to "Presence of Extra Points" without giving any explanation. Otherwise, if and only if the document discusses major additional points that are necessary to answer the question, output "True" and mention only the extra major points discussed.

Document:
{Document}

Question: {QUESTION}

Answer Points:
{ANSWER}

Give output in the following format:
Presence of Extra Points: True/False
Extra Points Mentioned (if any):

Figure 17: Programmatic prompt for verifying absence of relevant information in the documents for CHASE-QA.

---

**CHASE-QA: Verifying Presence of Answer**

**System Prompt:** You are an expert at verifying data.

---

You are given a document followed by a question and an answer point. You must check two things:
1. Presence: Is the point mentioned in the document?
2. Relevance: Is the point discussed in a manner such that it can be used to partially answer the question?

Document:
{DOCUMENT}

Question: {QUESTION}

Answer Point:
{ANSWER_POINT}

Give output in the following format:
Presence: True/False
Explanation for Presence:

Relevance: True/False
Explanation for Relevance:

---

Figure 18: Programmatic prompt for verifying presence of ground-truth answer in the documents for CHASE-QA.

---

**CHASE-QA: Solving the Task**

**System Prompt:** You are an expert at answering questions based on documents.

---

You are given some documents followed by a question. You need to generate the answer for that question. Provide the answer in bullet points, so that it is easier to objectively evaluate. Answering the question correctly requires information from multiple documents. You must only generate the points necessary for answering the question, without mentioning anything irrelevant to the question. If you find no relevant information in the documents for answering the question, you must only generate "No relevant information found in the documents." and nothing else.

Documents: {DOCUMENTS}

Question: {QUESTION}

Answer:

---

Figure 19: Programmatic prompt for solving examples in CHASE-QA.

---

**CHASE-QA: LLM-as-a-Judge for Calculating Accuracy**

**System Prompt:** You are an expert evaluator.

---

You are given a question, irrelevant answers, the ground-truth answer, and a prediction. You need to evaluate whether the prediction is correct by matching against the ground truth answer. Do not look for exact phrases or words since the prediction can have points that are a paraphrase of the same information. Based on the question, check for the presence of the same ideas or main points in the prediction as in the ground-truth answer. All the main points in the ground-truth answer must be mentioned in the prediction. The order of points mentioned is irrelevant. It is allowed for the prediction to elaborate or provide more specifics or details over the major points in the ground-truth answer. However, the prediction should not contain additional major points that are contradictory or irrelevant for answering the question. Importantly, the prediction must not discuss any of the points mentioned in the "irrelevant answers". The first word in your response must be either True or False. If False, explain why you think the prediction is wrong in detail.

Question: {QUESTION}

Irrelevant Answers: {IRRELEVANT_ANSWERS}

Ground-truth Answer: {GROUND_TRUTH_ANSWER}

Prediction: {PREDICTION}

Result:

Figure 20: Programmatic prompt for evaluating accuracy of predictions of models for problems in CHASE-QA.

---

**CHASE-QA: LLM-as-a-Judge for Calculating K-Precision**

**System Prompt:** You are an expert evaluator.

---

You are given a question, an answer written in points, and some documents. You need to check whether the information in the answer points is discussed in the documents in a manner such that it can be used to at least partially answer the question. You do not need to think about the overall correctness of the answer points, just check whether or not a particular answer point is discussed in the documents. Your goal is to calculate precision, i.e., the percentage (out of 100) of answer points that have been adequately mentioned in the document. The first thing in your response must be "Precision: " followed by the precision value in decimal form. If precision is less than 100%, explain which answer points are not present in the document.

Documents: {DOCUMENTS}

Question: {QUESTION}

Answer Points: {ANSWER_POINTS}

Result:

Figure 21: Programmatic prompt for evaluating K-Precision of predictions of models for problems in CHASE-QA.

**CHASE-QA: LLM-as-a-Judge for Calculating Recall**

**System Prompt:** You are an expert evaluator.

You are given a question, a statement, and some reference points. You need to check whether the information in the statement is discussed in the reference points in a manner such that it can be used to at least partially answer the question. It is okay if the reference points contain a lot more information, your goal is to only check whether the statement is included in the reference points. The first word in your response must be either True or False. If False, explain why in detail.

Question: {QUESTION}

Statement: {STATEMENT}

Reference Points: {REFERENCE_POINTS}

Result:

Figure 22: Programmatic prompt for evaluating recall of predictions of models for problems in CHASE-QA.

---

**CHASE-CODE: Generating Helper Functions**

**System Prompt:** You are an expert generator of code data.

---

You are a research scientist. You want to make data to test an advanced code generation system. You are given a domain. Assume that there is a large python code base 'C' with at least 10 python files on that domain.

Domain: {DOMAIN}

You need to create 5 functions in this codebase for achieving various objectives. First define the parameters that will be input to the function. Then define the objective of the function. The objective must consist of 3-4 sub-goals, each of which must involve complex logic that make it very difficult to implement the function. However, each sub-goal must be well-specified such that there is only one way to implement the sub-goal. Then based on the objective, you need to create a single function (do not create other functions inside this).

Some examples are:

Parameters:
- data: pandas.DataFrame
- k: int
Objectives:
- In the dataframe "data", find the "frequency" of occurence of rows that have at least one string field with the number of letters divisible by "k".

[redacted]

Function "filter_k_frequency" in file "string_filters.py":
```
1  import pandas as pd
2
3  def filter_k_frequency(data, k):
4      [redacted]
5      return frequency, filtered_df
```
Now you need to create 5 unique, diverse, and complex functions. Answer in the following format:

Function <Number>:
Parameters:
- <para_name>: <data_type>
...
Objectives:
- <sub_goal>
...

Function "function_name" in file "file_name.py":
<import statements>

<function definition>

Figure 23: Prompt for generating helper functions for CHASE-CODE.

---

**CHASE-CODE: Generating Problem Statement and Answer Code**

**System Prompt:** You are an expert generator of code data.

---

You are a research scientist. You want to make data to test an advanced code generation system.

Below, you are given 10 functions from a codebase "C" in the domain of {DOMAIN}.

Parameters:
- data: pandas.DataFrame
- k: int
Objectives:
- In the dataframe "data", find the "frequency" of occurence of rows that have at least one string field with the number of letters divisible by "k".

[redacted]

Function "filter_k_frequency" in file "string_filters.py":

```python
import pandas as pd

def filter_k_frequency(data, k):
    [redacted]
    return frequency, filtered_df
```

[redacted]

You need to create a complex function that calls at least 4 (but not more than 6) of these functions to achieve various objectives. Apart from just calling these functions, it should also implement some other pieces of complex logic. You first need to define the parameters that will be input to the function. Then you need to define the objective of the function. Follow these instructions for creating the objective:

1. The objective must consist of 6-8 sub-goals. Each sub-goal must be detailed and well-specified such that there is only one way to implement the sub-goal.
2. VERY IMPORTANT: The objective must not explicitly specify which functions should be called.
3. Always give names for variables you are talking about in the objective.
4. You must explicitly mention what parameters are to be used for a specific sub-goal by the name of the parameter.
5. Whenever a variable is obtained that must be returned by the function, you must explicitly state that in the sub-goal.
6. At least 2 of the sub-goals must involve some complex logic, apart from just calling helper functions that make it very difficult to implement the function.

Once you write down the objective, you need to create the function that achieves this objective. Import the required functions from the codebase "C" and use them in your function.

You must give output in the following format:

[redacted]

Figure 24: Programmatic prompt for generating problem statement and answer code for CHASE-CODE.

**CHASE-CODE: Generating Test Code**

**System Prompt:** You are an expert tester of code systems.

---

You are given a function. You need to define an input-output test case for that function to exhaustively test all scenarios.

{ANSWER_FUNCTION}

Follow these instructions:
1. You must output only a single long python code.
2. First initialize the input parameters for the function in python code. If the function reads data from files, you should create and store the necessary files with sample data in the corresponding filepath in the python code. Call the function and assign the return values to variables named as return_<variable_name>.
3. Then write new code to implement the exact logic of the function. This way, you need to simulate step-by-step how the values of the input parameters will be used to obtain the final return values. Call these values as correct_<variable_name>.
4. Finally, and most importantly use assert statements to compulsorily check if the returned outputs of the function (return_<variable_name> variables) match with the ones you computed yourself (correct_<variable_name> variables).

Give output in the following format:

```python
# Import statements if required
import <>
...

# Import function from file
from <filename> import <function_name>

# Initialize input parameters
<param1> = <value1>
...

# Call function with input parameters
return_<variable1>, return_<variable2>, ... = $<function_name>(<
    param1>, <param2>, ...)

# Step-by-step run-through of function to obtain intermediate
    outputs:

# Step 1
# Explanation: <>
<Code for step-1>

[redacted]

# Final Expected Output:
correct_<variable1> = <value1>
...

# Assert statements (compulsory) to check if the function returns
    the correct values:
assert return_<variable1> == correct_<variable1>
...
```

Figure 25: Programmatic prompt for generating the test code for CHASE-CODE.

---

**CHASE-CODE: Verifying if Function Executes**

**System Prompt:** You are an expert tester of code systems.

---

You are given a function in a file. You need to check whether the function correctly executes.

{FUNCTION}

Follow these instructions:
1. You must output only a single long python code.
2. First initialize the input parameters for the function in python code. If the function reads data from files, you should create and store the necessary files with sample data in the corresponding filepath in the python code.
3. Finally, call the function with the input parameters.

Give output in the following format:

```
 1 # Import statements if required
 2 import <>
 3 ...
 4
 5 # Import function from file
 6 from <filename> import <function_name>
 7
 8 # Initialize input parameters
 9 <param1> = <value1>
10 ...
11
12 # Call function with input parameters
13 return_<variable1>, return_<variable2>, ... = <function_name>(<
       param1>, <param2>, ...)
```

Figure 26: Programmatic prompt for generating the test code for verifying if a function executes correctly for CHASE-CODE.

---

**CHASE-CODE: Verifying Problem Statement**

**System Prompt:** You are an expert programmer.

---

You are given a codebase with some files and functions in the domain of params[0]. You need to write a single python function to achieve the objectives specified in the problem statement. You may call the functions in the codebase when necessary. Do not give any examples of usage or any explanations.

Codebase:

{RELEVANT_FUNCTIONS}

Problem Statement:

{PROBLEM_STATEMENT}

Give output in the following format:

```
1  # Import statements if required
2  import <>
3  ...
4
5  # Import necessary helper functions from their files
6  from <filename> import <function_name>
7
8  # Define the function
9  def <function_name>(<param1>, <param2>, ...):
10      # Your code here
11      ...
12
13      return <return_variable>
```

Figure 27: Programmatic prompt for verifying if the problem statement sufficiently specifies the answer code for CHASE-CODE.

**CHASE-CODE: Solving the Task**

**System Prompt:** You are an expert programmer. You must output only python code.

---

You are given a codebase. You need to write a single python function to achieve the objectives specified in the problem statement. In your function, you should call some of the functions in the codebase to achieve specific objectives. Do not give any examples of usage or any explanations.

Codebase:

{CODEBASE}

Problem Statement:

{PROBLEM_STATEMENT}

Give output in the following format:

```
1  # Import statements if required
2  import <>
3  ...
4
5  # Import necessary helper functions from their files
6  from <filename> import <function_name>
7
8  # Define the function
9  def <function_name>(<param1>, <param2>, ...):
10         # Your code here
11         ...
12
13         return <return_variable>
```

Figure 28: Programmatic prompt for solving examples in CHASE-CODE.

---

**CHASE-MATH: Breaking-down Seed MWPs**

**System Prompt:** You are an expert mathematician.

---

You are a research scientist. Your task is to create a hard math word problem to test an advanced math reasoning system. For that, you are given the following problem:

Q: {QUESTION}
A: {ANSWER}

Your job is to first divide up the problem into the "context" and the "question statement". Isolate the quantity that the problem is inquiring about by looking at the final question statement and the rest of the information provided becomes the context. Also form a brief answer statement by phrasing the answer in a complete sentence. Do not include the answer statement in the context.

Give output in the following format only:
Original context [without question statement]: <>
Question statement: <>
Original answer: <>
Original answer statement: <>

---

Figure 29: Programmatic prompt for breaking down the seed MWP for CHASE-MATH.

---

**CHASE-MATH: Creating Continuation of MWP**

**System Prompt:** You are an expert mathematician.

---

You are a research scientist. Your task is to create a hard math word problem to test an advanced math reasoning system. For that, you are given the following problem:

Context: {CONTEXT}
Question statement: {QUESTION_STATEMENT}
Answer: {ANSWER}
Answer statement: {ANSWER_STATEMENT}

You need to further continue the problem over the answer quantity, by introducing a scenario and new question where you need to perform one more operation (such as +,-,/,*, etc.) over this quantity to get the final answer. Crucially, the new context must not mention the original answer - it still has to be inferred based on previous information. Do not make any calculation or inference in the new context. Try to make the new context challenging. Also provide a complete reasoning of how you reached the new answer (never round down or round up decimals).

Give output in the following format only:
New operation over original answer: <>
New context [Do not mention original answer]: <>
New question statement: <>
New answer reasoning: <>
New answer [Number only]: <>

---

Figure 30: Programmatic prompt for extending the seed MWP for CHASE-MATH.

---

**CHASE-MATH: Solving MWP - 8-shot chain-of-thought**

**System Prompt:** You are an expert mathematician. Your final statement must be of the form 'The answer is <answer>'.

---

Solve the final math word problem given below by thinking step-by-step. You should always work with exact numbers - never round down or round up decimals based on context. Give the final answer in the end by saying "The answer is <number>".

Q: There are 15 trees in the grove. Grove workers will plant trees in the grove today. After they are done, there will be 21 trees. How many trees did the grove workers plant today?
A: There are 15 trees originally. Then there were 21 trees after some more were planted. So there must have been 21 - 15 = 6. The answer is 6.
Q: If there are 3 cars in the parking lot and 2 more cars arrive, how many cars are in the parking lot?
A: There are originally 3 cars. 2 more cars arrive. 3 + 2 = 5. The answer is 5.
Q: Leah had 32 chocolates and her sister had 42. If they ate 35, how many pieces do they have left in total?
A: Originally, Leah had 32 chocolates. Her sister had 42. So in total they had 32 + 42 = 74. After eating 35, they had 74 - 35 = 39. The answer is 39.
Q: Jason had 20 lollipops. He gave Denny some lollipops. Now Jason has 12 lollipops. How many lollipops did Jason give to Denny?
A: Jason started with 20 lollipops. Then he had 12 after giving some to Denny. So he gave Denny 20 - 12 = 8. The answer is 8.
Q: Shawn has five toys. For Christmas, he got two toys each from his mom and dad. How many toys does he have now?
A: Shawn started with 5 toys. If he got 2 toys each from his mom and dad, then that is 4 more toys. 5 + 4 = 9. The answer is 9.
Q: There were nine computers in the server room. Five more computers were installed each day, from monday to thursday. How many computers are now in the server room?
A: There were originally 9 computers. For each of 4 days, 5 more computers were added. So 5 * 4 = 20 computers were added. 9 + 20 is 29. The answer is 29.
Q: Michael had 58 golf balls. On tuesday, he lost 23 golf balls. On wednesday, he lost 2 more. How many golf balls did he have at the end of wednesday?
A: Michael started with 58 golf balls. After losing 23 on tuesday, he had 58 - 23 = 35. After losing 2 more, he had 35 - 2 = 33 golf balls. The answer is 33.
Q: Olivia has $23. She bought five bagels for $3 each. How much money does she have left?
A: Olivia had 23 dollars. 5 bagels for 3 dollars each will be 5 x 3 = 15 dollars. So she has 23 - 15 dollars left. 23 - 15 is 8. The answer is 8.
Q: {QUESTION}
A:

---

Figure 31: Programmatic prompt for solving an example in CHASE-MATH using chain-of-thought.

**CHASE-MATH: Solving MWP - 8-shot sentence-by-sentence**

**System Prompt:** You are an expert mathematician. Your final statement must be of the form 'The answer is <answer>'.

You need to solve the given math word problem. You should break down the problem sentence by sentence, and solve each sentence, one at a time, from start to finish until you get the final answer. You should always work with exact numbers - never round down or round up decimals based on context. Give the final answer in the end by saying "The answer is <number>".
Given below are illustrations of solving sentence-by-sentence:

Q: In a store, an Uno Giant Family Card costs $12. When Ivan bought ten pieces, he was given a discount of $2 for each. The store has a 8% sales tax added to all purchases. Ivan decides to save 25% of this expenditure for a future vacation. After saving, Ivan instead decides to split this amount between two of his friends who are always helping him out. One of the friends decided to split their received amount equally among the five children in Ivan's neighborhood who helped him move the previous day. How much will each child receive?

A: Sentence 1: Uno card costs $12.
Sentence 2: Ivan bought 10 cards and there was a discount of $2 each. So, 10 * $12 = $120 total cost of cards and, 10 * $2 = $20 discount.
Sentence 3: Sales tax is 8% of ($120 - $20 = $100). So, 8% of $100 = $8.
Sentence 4: Ivan saves 25% of ($100 + $8) = $108. So, 25% of $108 = $27.
Sentence 5: Ivan splits $27 between 2 friends. So, $27 / 2 = $13.50 each.
Sentence 6: One friend splits $13.50 among 5 children. So, $13.50 / 5 = $2.70 each.
The answer is 2.70.

[redacted]

Q; Carly is trying to get in shape to try out for the soccer team. She starts by running 2 miles a week. The second week, she runs twice as long plus 3 extra miles per week. The third week she runs 9/7 as much as she ran the second week. The week after that, she sprains her ankle and has to reduce her running time by 5 miles this week compared to the previous week. After a few weeks of recovering from her ankle injury, Carly starts to feel better and decides to gradually increase her running time. She starts with a shorter routine that is one quarter of the amount she ran the week she was injured. After a week of intense training, Carly decides to boost her speed and endurance by increasing her weekly running routine to 2.5 times longer. Carly is planning increase the weekly routine by 5 times now. How much is Carly planning to run every week?

A: Sentence 1: Nothing to solve.
Sentence 2: Carly runs 2 miles a week.
Sentence 3: Carly runs 2 * 2 + 3 = 7 miles in the second week.
Sentence 4: Carly runs 9/7 * 7 = 9 miles in the third week.
Sentence 5: Carly reduces her running time by 5 miles this week. So, 9 - 5 = 4 miles.
Sentence 6: Nothing to solve.
Sentence 7: Carly starts with 1/4 of 4 miles = 1 mile.
Sentence 8: Carly increases her running routine to 2.5 times longer. So, 1 * 2.5 = 2.5 miles.
Sentence 9: Carly wants to make her long run 5 times as long as her new routine. So, 2.5 * 5 = 12.5 miles.
The answer is 12.5.

Q: {QUESTION}
A:

Figure 32: Programmatic prompt for solving an example in CHASE-MATH sentence-by-sentence.

