# OpenReview forum: "How to Get Your LLM to Generate Challenging Problems for Evaluation"
_ICLR.cc/2025/Conference — Submitted to ICLR 2025_

### Official Review · Reviewer_yxKN · 2024-10-17

**Soundness:** 2
**Presentation:** 3
**Contribution:** 2
**Rating:** 3
**Confidence:** 4

**Summary:**

This paper introduces CHASE (CHallenging AI with Synthetic Evaluations), a framework for generating challenging evaluation benchmarks using large language models (LLMs). The authors implement CHASE to create benchmarks in three domains: document-based question answering, repository-level code completion, and math reasoning. Experiments with 15 LLMs show that the generated benchmarks are challenging, with even top models achieving only 40-60% accuracy across domains. The authors demonstrate CHASE's utility in differentiating between state-of-the-art models and revealing performance drops with increasing context length. They argue that this approach offers advantages in scalability, renewability, and ability to evaluate tasks difficult for humans to assess, while providing high-quality, challenging problems for LLM evaluation.

**Strengths:**

1. The experiments are comprehensive, with a good set of LLMs covering representative proprietary and open-source models.
2. The paper is well-written, which clearly describes the methods, experiments and results.

**Weaknesses:**

1. Although overall I believe it is valuable to explore data synthesis for benchmark construction, I think the authors should be more careful in selecting appropriate settings. I think the most important motivation for this paper is that it is expensive and sometimes impracticable to create benchmarks with challenging problems. However, in some settings present in the paper, I feel that this may not be the case. For example, SWE-bench [1] also focuses on repo-level code generation, and they take existing Github issues as queries, and the modifications made by real users as the ground truth. The current state-of-the-art performance is only 43% in the leaderboard, which indicates its difficulty. Compared to CHASE-CODE, I think the pipeline used in SWE-bench is a better way to collect repo-level code generation data.
2. To demonstrate that this pipeline is scalable, I think it is important to generate data of large size and apply it to training. If the API cost is a concern, I think the authors can use open-source models, e.g., Llama-70B.
3. Typo in Figure 1: Jill has 12 pens in the bottom right corner.
4. In line 443-444, I don’t quite understand why better performance of models different from the generator and verifier can indicate better data quality.
5. One advantage of the CHASE claimed by authors is to mitigate data contamination, but I think this may not be a big concern for challenging benchmarks that involve intensive reasoning. For example, even if codellama [2] has been intensively trained on Github data, its performance is still low on SWE-bench (which uses the Github data).

[1]. Jimenez, Carlos E., et al. "Swe-bench: Can language models resolve real-world github issues?." arXiv preprint arXiv:2310.06770 (2023). \
[2]. Roziere, Baptiste, et al. "Code llama: Open foundation models for code." arXiv preprint arXiv:2308.12950 (2023).

**Questions:**

1. How to organize functions into files to build repositories from scratch in CHASE-CODE?
2. Could you specify more details on rejection sampling?

---

> ### Author Response · Authors · 2024-11-15
> **Response to Weakness**
>
> Thank you for reviewing our paper. We are pleased to see that you found our results comprehensive and the paper to be well-written. Please find our response to specific comments below.
>
> **[W1]** *SWE-bench [1] also focuses on repo-level code generation*
>
> We emphasize that our main contribution is the end-to-end framework for generating challenging synthetic data for evaluation, and we show how it can be applied in the code generation domain.
>
> Our goal is not to present a code generation benchmark that competes with benchmarks like SWE-Bench. Our generation process is a complementary method to SWE-Bench's way of creation. Moreover, it is reasonable to assume that models will eventually become capable at SWE-Bench in the near-future (either due to contamination or due to genuine progress). Since such high-quality data curation will become increasingly difficult to do manually, it is very important to explore alternatives, including synthetic data evaluation strategies. Our method (and possibly future improvements) can automatically generate hard data that is challenging for even the most capable models themselves. Moreover CHASE facilitates a much higher level of controllability, i.e., we can generate the specific types/domains of code that we want to evaluate or analyze (like we did for algorithms vs data-preprocessing) and it is not bottlenecked by the availability of high-quality repositories with exhaustive tests.
>
> **[W5]** *data contamination not a big concern for challenging benchmarks… even if codellama [2] has been intensively trained on Github data, its performance is still low on SWE-bench.*
>
> We would like to note that the Codellama paper [2] clearly mentions it has not been trained on meta-level or temporal information such as issues or commits. As for the other SOTA models, they do not disclose their data so it is hard to comment on whether or not SWE-bench data is already a part of their training set.
>
> We respectfully disagree that data contamination is not a big concern. There is significant evidence to suggest that models are showing improved accuracies at benchmarks like GSM8k [3] because of contamination [4][5].
>
> In contrast, the contexts and problems of datasets created using CHASE are completely novel for all models, which makes for a better test of generalization. Moreover, if and when these datasets get saturated/contaminated, new test suites can be sampled from the (more powerful) LLMs of the future (possibly with improved iterations of our pipeline).
>
> Lastly, note that the SWE-Bench idea is relatively new (<1 year old), so it is possible that leading AI companies have not yet incorporated this type of meta-data in their training set. However, it is reasonable to expect they will do so, in a format similar to the one used in SWE-Bench, to train future iterations of  LLMs, which will then lead to a much higher performance.
>
> **[W2]** *To demonstrate that this pipeline is scalable, I think it is important to generate data of large size and apply it to training*
>
> We respectfully disagree and it is a mischaracterization of our work. Our focus in this work is to obtain challenging data for evaluation. We have already generated a significant amount of data (our new experiments more than doubled the amount of data we generated earlier). We can generate more if needed, but generation is not cheap, and we believe that what we have generated so far is sufficient for challenging the currently available LLMs.
>
> Further note that we indeed generated ~10k math problems using Llama-3.1-8B for our finetuning experiments (L486-493). Hence, we have demonstrated that our approach is clearly scalable.
>
> **[W4]** *why better performance of models different from the generator and verifier can indicate better data quality.*
>
> Our point with this statement was to suggest that the generated data is not too biased towards the generator or the verifier since other models are also performing better. However, this is a minor point, and we are willing to remove it if it is still confusing.
>
> We hope we made the focus and scope of our work clear and clarified potential misunderstandings. We kindly ask you to consider raising the score.
>
> References:
>
> [1] Jimenez et al. (2024). SWE-bench: Can Language Models Resolve Real-World GitHub Issues? In ICLR.
>
> [2] Rozière et al. (2023). Code Llama: Open Foundation Models for Code. In Arxiv: 2308.12950.
>
> [3] Cobbe et al. (2021). Training Verifiers to Solve Math Word Problems. In Arxiv:2110.14168.
>
> [4] Zhang et al. (2024). A Careful Examination of Large Language Model Performance on Grade School Arithmetic. In Arxiv:2405.00332.
>
> [5] Mirzadeh et al. (2024). GSM-Symbolic: Understanding the Limitations of Mathematical Reasoning in Large Language Models. In Arxiv:2410.05229.

---

> ### Author Response · Authors · 2024-11-15
> **Response to Questions**
>
> **[Q1]** *How to organize functions into files to build repositories from scratch in CHASE-CODE?*
>
> We randomly sample irrelevant helper functions and combine them with the relevant helper functions for a particular problem. The ordering of these functions is then randomly permuted and distributed among 10 python files.
>
> **[Q2]** *Could you specify more details on rejection sampling?*
>
> We run GPT-4o-mini twice (with temperatures 0.3 and 0.7) on the generated problems. Depending on the difficulty of the task, we remove a percentage of problems that GPT-4o-mini answered correctly on both runs. The reasoning is that we believe such problems will be easy to solve for most of the SOTA models as well and therefore we decrease their population in the final dataset to yield a challenging benchmark with lots of room for improvement.

---

> ### Comment · Reviewer_yxKN · 2024-11-21
> **Response**
>
> I think the efforts to curate SWE-bench are not very significant, as they use existing Github issues, repos and verified commits with test cases. The reason I mention this dataset is not to ask for a better quality of generated benchmark but to seek stronger motivations about why we want to use a synthesized code generation benchmark if a very realistic one already exists. I feel like the way to create SWE-bench is scalable, e.g., they can efficiently collect a more difficult one using the same pipeline in other repos. The evaluation on such benchmarks will be the most accurate to reflect how capable LLMs are in assisting programmers in software engineering tasks.

---

> > ### Author Response · Authors · 2024-11-21
> > **Response to Reviewer**
> >
> > Respectfully, your argument is not valid.
> >
> > First, if we assume that SWE-Bench is scalable (it is not as we will explain later), CHASE-Code is still a different paradigm as explained below:
> >
> > **Task.** SWE-Bench focuses on a different task compared to CHASE-Code. The task in SWE-Bench is about code editing/fixing or generating patches for “issues” or “bugs” in a repository. The task in CHASE-Code is about generating new functionality based on its precise description. Such problems may be there in SWE-Bench, but they are a clear minority (see Table 13 in [1]). You can also qualitatively compare the types of problem statements in SWE-Bench and CHASE-Code.
> >
> > **Controllability.** A very important feature of CHASE-Code is that you can control the parameters of the problem you want to design. You can choose the domain (such as algorithms, data pre-processing, etc.), how complex you want the function to be, what length the repository should be, etc. This is not possible for SWE-Bench.
> >
> > Hence, **the scalability or existence of a high-quality SWE-Bench does not decrease the impact of CHASE-Code.**
> >
> > Now, we motivate why synthetic data for code generation is a good avenue to pursue. We have already explained above that synthetic data generation provides a **high level of controllability**. Further, benchmarks like SWE-Bench are not highly scalable. They are bottlenecked by the availability of well-maintained, high-quality repositories with extensive tests for each issue (which is why they focused on 12 popular repositories). In contrast, synthetic data generation allows for automatic creation of tests (as shown in CHASE-Code), which makes it **comparatively more scalable**. Lastly, note that SWE-Bench is a good way to collect challenging data for one particular type of task. There are many other tasks in the domain of code (such as the task in CHASE-Code, the task of code understanding [2], competitive programming [3], etc.) where there may not exist good ways of curating challenging data without human intervention.
> >
> > References:
> >
> > [1] Jimenez et al. (2024). SWE-bench: Can Language Models Resolve Real-World GitHub Issues? In ICLR.
> >
> > [2] Gu et al. (2024). CRUXEval: A Benchmark for Code Reasoning, Understanding and Execution. In Arxiv:2401.03065.
> >
> > [3] Li et al. (2022). Competition-Level Code Generation with AlphaCode. In Arxiv:2203.07814.

---

> > > ### Comment · Reviewer_yxKN · 2024-11-24
> > > **Thanks a lot for the reply**
> > >
> > > First of all, I am grateful to the opportunity to engage in these meaningful discussions with the authors.
> > >
> > > The CHASE-Code focuses on generating new functionality based on its precise description. In the paper, the description is "Given a repository of Python functions, the task is to implement a new function based on a set of objectives provided in natural language". If I understand it correctly, this seems to be the only type of code generation problem considered in the paper. Additionally, the data only covers two domains (which may not be regarded as "broad"). I think the synthetic benchmark will be more valuable if it is large, diverse and comprehensive as the main claim here is that it requires much less human efforts, so should be very easy to scale up. The current version with only 220 problems in the dataset seems to be also practical for human annotators to finish in a reasonable period.
> > >
> > > Based on the example presented in the middle of Figure 2, I recognize that the natural language description is very detailed. I think there is a difference between precise/clear and detailed instruction. In realistic scenarios, it is not common to see a code generation problem specified in such a detailed manner (It is so detailed that even which specific parameter (price_col in Figure 2) to focus is described). In LeetCode, the objective describes the target goals instead of giving implementation details, while in realistic scenarios to solve an issue in Github repo, users or LLMs even need to figure out which file or while function to modify. I think these will reflect more of the genuine needs for code generation.
> > >
> > > Regarding controllability, I recognize the good motivation of authors. However, I believe that the authors need to find better scenarios to demonstrate it. I think a good scenario people would want to use the synthesized benchmark is: (1). there is very rare existing data to leverage; (2). The scenario is realistic and important. Specific to code generation, I think it is not very difficult to find problems in the domain data pre-processing and algorithms from Kaggle, LeetCode, codeforces, etc. If the author can find good scenarios and control the data generation on them, I think it will make more sense to me.
> > >
> > > Hope these address the author's questions.

---

> > > > ### Author Response · Authors · 2024-11-24
> > > > **Response to Reviewer**
> > > >
> > > > Perhaps you missed our new experiments as mentioned in our responses above: CHASE-Code now has 500 problems. This data size is adequate for a long-context code generation benchmark. Contemporary long-context benchmarks [1,2,3] have only ~500 examples per task and even widely used code benchmarks like HumanEval [4] have ~160 examples. We have shown that our approach is highly scalable. If you disagree, then we ask you to point out concrete bottlenecks with our approach to explain why it is not scalable.
> > > >
> > > > *The current version with only 220 problems in the dataset seems to be also practical for human annotators to finish in a reasonable period.*
> > > >
> > > > It would be helpful if the reviewer provides evidence for this statement. As far as we are aware, there is no repository-level code generation benchmark of such high difficulty that can be created at similar expense (time and cost) as CHASE-Code (see Table 5 in Appendix). We have already explained how SWE-Bench is a different paradigm, so it is illogical to compare against it.
> > > >
> > > > *In realistic scenarios, it is not common to see a code generation problem specified in such a detailed manner*
> > > >
> > > > We would like to understand your exact concern here. Yes, CHASE-Code perhaps simulates a scenario which is a bit easier than completely realistic scenarios. But still it is difficult for LLMs to solve, and hence it is a valuable benchmark. If you believe there is no value in such slightly-less-realistic benchmarks, then you would have to consider almost all benchmarks in NLP so far to be valueless.
> > > >
> > > > *I think it is not very difficult to find problems in the domain data pre-processing and algorithms from Kaggle, LeetCode, codeforces, etc.*
> > > >
> > > > This is an unfair characterization of our work. We target repository-level code completion problems. None of these sources have any repository-level code.
> > > >
> > > > We request you to precisely state your standing concerns with the paper. It is our opinion that **a disagreement over the “realness” of the scenario of just one example benchmark generated using our framework does not merit a score as low as 3**. We remind you that **our main contribution is a general framework to create challenging synthetic data for evaluation across multiple domains**. This is the **first time this problem is being studied**. We have also shown its applicability for two other scenarios such as document-based question answering and math reasoning.
> > > >
> > > > Our claim in this paper is that generating challenging data for evaluation using humans could be difficult or impractical for many reasons (cost, expertise, long-context generation, etc). Hence, we need to study the problem of generating synthetic data for evaluation. We have presented a general framework to do this, and showed its applicability across multiple domains. Even if you disagree with the exact way this was done for one particular domain (and note that we have provided concrete arguments in our responses for this), do you still think that this paper is not a valuable contribution?
> > > >
> > > > References:
> > > >
> > > > [1] Zhang et al. (2024). ∞Bench: Extending Long Context Evaluation Beyond 100K Tokens. In ACL.
> > > >
> > > > [2] Li et al. (2024). LooGLE: Can Long-Context Language Models Understand Long Contexts? In ACL.
> > > >
> > > > [3] Wang et al. (2024). Leave No Document Behind: Benchmarking Long-Context LLMs with Extended Multi-Doc QA. In EMNLP.
> > > >
> > > > [4] Chen et al. (2021). Evaluating Large Language Models Trained on Code. In Arxiv:2107.03374

---

> ### Comment · Reviewer_yxKN · 2024-11-25
> **Response**
>
> 1. I think the dataset size with 220 or 500 examples is too small to demonstrate the value of synthesized benchmark, because it is not even larger than many human-curated benchmarks. I think authors need to provide more justifications on why it will be difficult or time-consuming for the synthesized dataset to be larger.
> 2. If the synthetic dataset does not reflect realistic scenarios, then I think the authors need to justify the motivation about why we need such a benchmark, or in what cases people will want to synthesize a benchmark, etc.
> 3. To find existing resources on data-preprocessing or algorithms, I think issues in repos like pandas or numpy will provide related problems. Overall, I think it would be more valuable to demonstrate that constructing benchmarks with existing resources is difficult, so we need to rely on synthetic data.
>
> Additionally, although I may have different opinions from the authors, I hope that the author's response could be less aggressive (although this is very minor).

---

> ### Author Response · Authors · 2024-11-25
> **Response**
>
> We have provided concrete arguments against your concerns. We feel you are just repeating your concerns again without pointing out any flaws in our response. We shall again provide counter-arguments for your points.
>
> **Data Size.** We don’t generate more data out of choice, and not just because of the cost. Indeed the cost and speed of generation would be much faster compared to humans (see Table 5). We made this choice because we want to keep our datasets accessible for other researchers (even currently, it costs $50 to run a single model inference once). The **value in our synthetic benchmark does not come from the quantity of examples, rather the value that we want to emphasize is that we are 'automatically' able to create 'challenging' problems, irrespective of the scale**. And considering precedence of other benchmarks as cited in our previous response, this scale is sufficient to serve for evaluation. Lastly, note that the **CHASE approach by design is quite easily scalable** (as we even showed by quickly scaling from 220 to 500 examples in CHASE-Code and generating 10k examples for the fine-tuning experiments). Its just that we didn't feel the need to scale it too much for these evaluations.
>
> **Regarding point 2.** We have provided ample arguments in defence of our code generation scenario. Our scenario is “repository-level code completion” with “complete specification of desired functionality”. Yes, we agree complete specification is hard to find in real scenarios, but it is still objectively an easier task to accomplish compared to its more real counterpart (less-specific problem statement) and we have shown LLMs still fail on it. This makes our benchmark an essential step towards evaluating the completely realistic scenario.
>
> **Regarding point 3.** You are still talking about data collection with GitHub “issues”. We have already explained we focus on a different code generation task: that of generating new code functionality based on user intent or problem statement. Moreover, we ask how you would get example-specific test code? We have already explained, obtaining tests is a huge bottleneck in "scraping-existing-code" creation approaches. This is not the case for our approach.
>
> Perhaps to better understand our motivation, we ask you to think about this realistic task: “A user is working on a codebase. They now want to implement a new functionality. They specify a description of the functionality to the LLM and the LLM generates the corresponding code”. How would you create a benchmark to evaluate this task? Note that this is significantly different from the SWE-Bench task which focuses on fixing issues and bugs. Human annotation for such tasks is expensive and requires high expertise. With CHASE-Code, we simulate this scenario, only with better-specification (which is an easier task).
>
> We again remind you that your concerns (that we have provided arguments for) are limited to CHASE-Code, making your score quite disproportionate. Our main contribution is the general framework to create challenging synthetic data for evaluation across multiple domains. This is the first time this problem is being studied. We have also shown its applicability for two other scenarios such as document-based question answering and math reasoning.
>
> We are not troubled by different opinions, indeed we welcome constructive criticism. However, we request you to kindly take stock of our arguments and attempt to refute them if you are still dissatisfied.

---

### Official Review · Reviewer_gKcy · 2024-11-01

**Soundness:** 3
**Presentation:** 3
**Contribution:** 2
**Rating:** 3
**Confidence:** 4

**Summary:**

To address the high cost of manually labeled data and the low quality of synthetic data, this paper proposes the CHASE framework. CHASE is a framework for automatically generating QA pairs, code problems, and math problems. It adopts an innovative bottom-up structure and divides the overall task into individually verifiable sub-tasks. This allows a seed problem to progressively increase in difficulty through multiple rounds of generation and verification. Experimental results show that the data generated by the CHASE have a certain degree of difficulty.

**Strengths:**

- The problem addressed by this paper is critical to the evaluation of current LLMs -- the lack of comprehensive and challenging datasets.
- The paper is well-structured, with comprehensive appendices, such as a detailed list of prompts used in CHASE.
- This paper presents a novel paradigm for data construction, which may have significant potential in the field of synthetic data.

**Weaknesses:**

- Some issues with the details of the paper. For example, in the main figure (Figure 1), the bottom-right corner should say "12 pens" instead of "18 pens."
- The current dataset is relatively small, which may result in a high degree of randomness in evaluation results when using this dataset.
- The experiments are not sufficiently thorough. Some experimental designs lack strong motivation, and there is a lack of experiments that demonstrate the advantage of CHASE over other synthetic data generation methods.

**Questions:**

1. Why does Figure 1 only provide an overview of constructing CHASE-QA and CHASE-Math, but not CHASE-Code? I believe all three should be at the same hierarchical level.
2. Without human verification, how can we ensure that the data in the CHASE-QA, CHASE-Code, and CHASE-Math datasets are correct? Is there a possibility that the ground truth/golden answer in the dataset themselves are incorrect?
3. In lines 333-340, you mentioned that approximately 33% of the data was filtered out through sampling and self-consistency, and subsequent experiments (e.g., Table 2) suggest that CHASE-QA generates more challenging data. I think it is unconvincing. If the 33% of the data were added back, how would the experimental results change? Would you still claim that CHASE-QA is a more challenging dataset?
4. From the examples given in the paper, CHASE-Math seems to concatenate a series of atomic problems. Intuitively, if the tested LLMs reason and calculate sentence by sentence, the accuracy may be significantly higher than under your current naive prompt. Could you elaborate further on how CHASE-Math is more challenging, given the point I raised?
5. What is the motivation behind the experiments in lines 469-477 and lines 486-493? In my understanding, the "Impact of context size" is not the focus of this paper. Also, the experiment in lines 486-493 only fine-tuned weaker models. Would the same conclusion apply to fine-tuning stronger models?
6. Could you provide some comparative experiments between the CHASE dataset and other synthetic datasets, such as a comparison between CHASE-QA and existing long-context benchmarks?

---

> ### Author Response · Authors · 2024-11-15
> **Response to Weakness and Questions**
>
> Thank you for reviewing our paper. We are glad that you found our research problem critical to study and our approach novel and innovative. Please find our response to specific comments below.
>
> **[W2]** *The current dataset is relatively small*
>
> We have carried out new experiments that have significantly scaled the size of the data. We now have 500 problems for CHASE-Code, and 500 problems for CHASE-Math (more than twice compared to before). We believe these sizes are sufficient to draw conclusions from our experiments. Further note that CHASE-QA and CHASE-Code are long-context benchmarks, and it would be prohibitively expensive (making them less accessible for other researchers) to test models on them if they contain too many examples.
>
> It is also important to note that the main contribution of this work is an end-to-end framework (i.e., the CHASE method) that can be used to automatically generate as much data as needed
>
> **[W3]** *Some experimental designs lack strong motivation*
>
> We have addressed this in the answer to Q5 below.
>
> **[W3]** *there is a lack of experiments that demonstrate the advantage of CHASE over other synthetic data generation methods*
>
> We would like to highlight that there exists no comprehensive framework to generate synthetic data for evaluation (for detailed discussion, please check our related works section). While there are many pipelines for generating synthetic data for training, they offer no simultaneous solution to the 2 core problems when generating data for evaluation (which our approach targets) - difficulty (for the generating LLM itself) and automatic verification. We did indeed compare CHASE with two popular synthetic data generation pipelines - self-instruct [1] and evol-instruct [2] (see L457-467 and Table 2), and concretely show the benefits of CHASE along the aforementioned dimensions.
>
> **[Q1]** *Figure for CHASE-Code.*
>
> We felt it was redundant and that it would clutter the main figure. We have now provided the figure in the appendix (see Fig. 4 on page 20).
>
> **[Q2]** *how can we ensure that the data in the CHASE-QA, CHASE-Code, and CHASE-Math datasets are correct?*
>
> We have manually verified each data point in CHASE-Math. It is impractical to manually verify each example in CHASE-QA and CHASE-Code because the context length for each example is 5-20k tokens. Further CHASE-Code requires a high level of technical expertise for verification. For these reasons, we randomly sampled 30 examples from CHASE-QA and CHASE-Code each, and manually verified them ourselves (discussed in L524-L533), which gives us a high level of confidence about the correctness of the data. To put in context the impracticality of manual verification, it took the author over 10 hrs to verify these 60 examples.
>
> **[Q3]** *you mentioned that approximately 33% of the data was filtered out…Would you still claim that CHASE-QA is a more challenging dataset?*
>
> We are simply discarding a portion of the generated data that we know can be easily solved even by weaker models. Including such “easy” examples does not serve much purpose in an evaluation benchmark if most models are going to be able to solve them. Indeed, our goal in this paper is to automatically find challenging problems that LLMs will struggle to solve, so we prioritize difficulty of problems over quantity. If we added those examples back, we expect the accuracy to be higher. However, ideally, we would like our method to create benchmarks where the performance of models reflects as much room for improvement as possible.
>
> In reference to the experiments in Table 2, note that the same type of filtration was carried out for the baselines as was done for CHASE (we have now made this explicit in the paper: L462). Hence, yes we would still conclude CHASE-QA is more challenging (apart from having much higher quality).
>
> **[Q4]** *Intuitively, if the tested LLMs reason and calculate sentence by sentence, the accuracy may be significantly higher than under your current naive prompt.*
>
> Intuitively, we agree, and this is how humans might reason and perform well on this benchmark too. However, LLMs are still prone to various kinds of mistakes. We experimented with a new prompt following your suggested intuition (see section C.1, Fig. 29 and Table 5). While the performance of models does increase by ~3%, the task is still very challenging for the models. We have also provided an example of an error made by Gemini-1.5-Pro under this new prompt (see Fig. 8) - the model solves perfectly till sentence 6, but then forgets that it has to use the previously calculated value for the next step.

---

> ### Author Response · Authors · 2024-11-15
> **Response to Questions (Cont'd)**
>
> **[Q5]** *What is the motivation behind the experiments in lines 469-477 and lines 486-493?*
>
> The motivation for the context size experiments is to show how we can synthetically add irrelevant context information to evaluation examples to make them even more challenging for LLMs to handle. These results highlight one particular dimension, i.e., “context-size”, which can be very easily controlled in synthetic data generation pipelines to craft challenging problems.
>
> The motivation for the fine-tuning experiments is to show that while smaller models such as Llama-3.1-8B can use CHASE to generate useful training data, they still perform poorly on CHASE data generated by a much more powerful model. Therefore, smaller models cannot “hack” the benchmark just by knowing the recipe of generation.
>
> **[Q6]** *Could you provide some comparative experiments between the CHASE dataset and other synthetic datasets, such as a comparison between CHASE-QA and existing long-context benchmarks?*
>
> Our main contribution is the CHASE methodology for generating difficult evaluation data. The byproducts are the resulting datasets. Our emphasis is more on the methodology contribution. So we compared with two other popular synthetic data generation methods -- Self-instruct [1] and Evol-instruct [2]  -- and found CHASE to generate superior data.
>
> Regarding comparison of CHASE-QA with other long-context benchmarks, we have added a detailed discussion in Appendix D.3 (L1142-L1163). But note that these are manually-annotated datasets. We are not aware of any synthetic long-context QA benchmarks targeting realistic scenarios and kindly request the reviewer to point to appropriate references and elaborate more on what kind of comparative experiments they hope to see.
>
> We have carried out new experiments and clarified the writing in multiple places in response to your review. We kindly ask you to consider raising your score if your concerns are addressed.
>
> References:
>
> [1] Wang et al. (2023). Self-Instruct: Aligning Language Models with Self-Generated Instructions. In ACL.
>
> [2] Xu et al. (2024). WizardLM: Empowering Large Pre-Trained Language Models to Follow Complex Instructions. In ICLR.

---

> > ### Comment · Reviewer_gKcy · 2024-11-18
> > **Response to Authors**
> >
> > I appreciate your detailed and patient response. I have carefully reviewed your revised paper (which I noticed mainly added content to the appendix) as well as your discussions with other reviewers. Below are my comments:
> >
> > **Regarding your response to [W2]:** First, I understand that the primary contribution of this work is the bottom-up automated framework for constructing challenging problems. However, I am not the only reviewer who raised concerns about the dataset size. Current mainstream benchmarks typically consist of at least a few thousand examples. As for the cost of generating more data, I believe that this is an important factor that an effective and broadly applicable framework should take into account. From the users’ perspective, both the time and financial cost of constructing data are clearly essential considerations (I would also like to see experiments related to this aspect). This is especially relevant since the primary area of this paper is **"datasets and benchmarks"** (which also relates to your statement, "The byproducts are the resulting datasets." In my opinion, the dataset itself should still be one of the major contributions in this track).
> >
> > **Regarding your response to [W3]:** You mentioned that "there exists no comprehensive framework..." I believe that since the title of the paper highlights “Challenging Problems,” it would be helpful if you compared the difficulty of the data generated by CHASE with that of other similar datasets. This would allow us to better understand the quality of CHASE dataset. This is unrelated to whether the process is fully automated or whether it addresses the “simultaneous solution to the 2 core problems”. Furthermore, I think that your framework does not "offer a simultaneous solution to the two core problems," since the generation process for QA, code, and math datasets differs in terms of prompts and procedures. The only commonalities among them seem to be the use of a LLM generator, a verifier, and the "bottom-up" concept. I may have misunderstood, so I welcome your clarification.
> >
> > **Regarding your response to [Q2]:** For the manually verified results, you mentioned that 6%-7% of the data is incorrect. In extreme cases, this means that the accuracy range for LLMs evaluated with CHASE could be off by as much as ±7%, which may be unacceptable in the current LLM evaluation landscape. This is especially concerning, as newly released models often outperform current SOTA LLMs by only 1%-2% on some evaluation sets.
> >
> > **Regarding your response to [Q3]:** It seems you may have misunderstood my question. My concern is that the experiments presented in Table 2 aim to demonstrate that the CHASE dataset produces more challenging problems than direct prompting approaches, as you mentioned in your response to [W3]. Therefore, I believe it is unfair to filter out easy examples from the CHASE dataset and then compare it with direct prompting approaches. At the very least, this comparison should either include or exclude filtering for both approaches to ensure a fair comparison.
> >
> > **Regarding your response to [Q4]:** I found the example in Figure 8 inappropriate. The sentence "He decides to continue running but at double the distance he covered during his recovery week for each day the next week, aiming to improve his overall performance.” is highly ambiguous. I tested it, and if the sentence is changed to "The distance he ran each day was twice the total distance he ran during the recovery week," Gemini-1.5-Pro can answer it correctly. This makes me question whether there are many similar issues in the CHASE-Math dataset.
> >
> > **Regarding your response to [Q5]:** First, if simply increasing the context size can make the problem much more challenging, then how does this demonstrate the significance of CHASE? For example, you should conduct experiments comparing the performance of LLMs on CHASE problems with irrelevant context added, versus regular (seed) problems with irrelevant context added, to observe the trends in accuracy. Furthermore, the experiment with fine-tuning smaller LLMs seems insufficiently rigorous. You cannot conclusively state that stronger open-source models (e.g., llama3-70B) would not be able to "hack" the benchmark, as the success of hacking may depends non the strength of the LLM.
> >
> > In conclusion, I appreciate your efforts to address some of my concerns. However, not all issues have been resolved, and thus **I am unable to raise my score at this time**.

---

> > > ### Author Response · Authors · 2024-11-21
> > > **Response to Reviewer**
> > >
> > > We would like to thank you for engaging in the discussion.
> > >
> > > **Size of data.** We may be mistaken but it seems you are the only reviewer with concerns about the data size (reviewer yxKN’s point is about scaling for training, which we have separately addressed in our response to them). While it is true that many contemporary benchmarks have a few thousand examples, there are also many benchmarks that have much less examples. For instance, HumanEval [1] and SVAMP [2] are widely used benchmarks for code and math reasoning that have ~160 and ~1000 examples respectively. We would also like to point out example papers (whose main contribution is the dataset) such as EvoCodeBench [3], a repo-level code generation benchmark, and MuSR [4], a story-based QA benchmark published at ICLR and NeurIPS this year that have ~275 and ~750 examples respectively. Further note that many contemporary long-context benchmarks [5,6,7] have only ~500 examples per task. Hence, we believe that the amount of data we generated is sufficient for benchmarking current LLMs and for supporting our claims and conclusions. We would also like to note that we are not too troubled by the cost of generation (we added Table 5 and L1075-1079 providing the costs of generation). However, we also wish to keep our long-context benchmarks accessible to play around with for researchers with limited resources (currently it costs ~$50 to run inference for just one SOTA model on our benchmark).
> > >
> > > **Comparing difficulty of data.** We have provided model performance accuracies for our benchmarks, which sufficiently show its difficulty. We have now added discussions comparing the performance of models on our datasets against other widely-used challenging benchmarks in those domains in Section C.1 and Tables 6 and 7. If you would like more specific analysis, could you let us know how?
> > >
> > > **Correctness of problems.** We would like to highlight that this issue persists mostly in CHASE-QA, because we couldn’t find errors in CHASE-Code on inspection, and we have filtered out incorrect examples from CHASE-Math. We agree that this is a limitation, but it comes with the benefit of providing a very real-world test scenario. To give more context, the errors in CHASE-QA pertain to the presence of extra relevant information in the documents which is not mentioned in the ground-truth (see Fig. 9 for an example). We have now also included evaluation with softer metrics, such as ‘K-Precision’, which measures the how faithful the model prediction is to the given documents, and ‘Recall’, which measures whether the model prediction includes all the information in the ground-truth (while allowing the model to provide more information). The results are discussed in Appendix C.2 and Table 8. Note that the gaps in performance between many models on CHASE-QA (for both, accuracy and recall) is quite large, which does make the conclusions we draw to be valid. We have now also manually reviewed 30 examples from the QA dataset generated by the direct generation baseline and found that it had 9 errors (~30%). We had already reported 34% error for the math problems generated by the direct generation baseline. This shows the advantages of CHASE in generating more correct problems.
> > >
> > > **Regarding Q3.** We believe we did answer this point in our response above. We have carried out filtration for both, CHASE and the direct generation baselines, which makes the comparison fair.
> > >
> > > **Regarding Q4.** It is difficult for us to understand why the statement is ambiguous. There is no notion of distance covered per day in the recovery week, so the most probable meaning of “distance he covered during his recovery week” has to be the total distance in the recovery week. Indeed, if we just add “total” in front of the “recovery week”, the model still makes the same mistake. In any case, we present another failure case for you to look at in Figure 9.
> > >
> > > References:
> > >
> > > [1] Chen et al. (2021). Evaluating Large Language Models Trained on Code. In Arxiv:2107.03374
> > >
> > > [2] Patel et al. (2021). Are NLP models really able to solve simple math word problems? In NAACL.
> > >
> > > [3] Li et al. (2024). EvoCodeBench: An Evolving Code Generation Benchmark with Domain-Specific Evaluations. In NeurIPS Datasets and Benchmarks Track.
> > >
> > > [4] Sprague et al. (2024). MuSR: Testing the Limits of Chain-of-thought with Multistep Soft Reasoning. In ICLR.
> > >
> > > [5] Zhang et al. (2024). ∞Bench: Extending Long Context Evaluation Beyond 100K Tokens. In ACL.
> > >
> > > [6] Li et al. (2024). LooGLE: Can Long-Context Language Models Understand Long Contexts? In ACL.
> > >
> > > [7] Wang et al. (2024). Leave No Document Behind: Benchmarking Long-Context LLMs with Extended Multi-Doc QA. In EMNLP.

---

> > > > ### Author Response · Authors · 2024-11-21
> > > > **Response to Reviewer (Cont'd)**
> > > >
> > > > **Regarding Context Size.** Perhaps you have misunderstood our approach. For the long-context domains, i.e., QA and Code, we do not start with any seed example. The examples are generated completely from scratch. Without the CHASE framework, you will not be able to craft a valid example to add irrelevant context to. Note that the context size is just one dimension of difficulty, there are many others (considering the low accuracies we see even without adding irrelevant context). Further note that there is no element of difficulty arising from context size in CHASE-Math.
> > > >
> > > > **Regarding Finetuning.** Note that our claim is only meant for much weaker models (we have made this more clear in the text now). We only wished to focus on fine-tuning with smaller models of ~7B scale since that can be done more accessibly. Finetuning larger models introduces a lot more complexity and is currently outside the scope of this work, which primarily focuses on evaluation.

---

> ### Author Response · Authors · 2024-11-25
> **Request to engage in discussion**
>
> We kindly ask you to increase your score if your concerns are addressed by our responses and additional experiments. There are only 2 days left in the discussion period.

---

> ### Author Response · Authors · 2024-11-29
> **Request to engage in discussion**
>
> Dear Reviewer,
>
> We request you to kindly read our response and adjust your score if your concerns are addressed.

---

### Official Review · Reviewer_CAvC · 2024-11-03

**Soundness:** 3
**Presentation:** 3
**Contribution:** 3
**Rating:** 6
**Confidence:** 4

**Summary:**

The authors introduce CHASE, a unified framework to synthetically generate challenging problems
using LLMs without human involvement. For a task that is given, the approach builds a hard problem
 in a bottom-up manner from simpler components. It decomposes the generation process into
independently verifiable sub-tasks to ensure a high level of quality and correctness.

CHASE is designed to address two challenges that the authors state succinctly on pages 1 and 2:
first, how can it be used to create hard and realistic problems, and secondly, how can it be used
to automatically verify the correctness of the generated data? This second challenge is especially
prevalent in other work of this nature that is attempting to construct synthetic evaluation benchmarks.

**Strengths:**

--The paper is written at an impressive quality, especially the figures and the elucidation of the problem's motivation and challenges.
--The authors consider three sufficiently diverse tasks and benchmarks to showcase the utility of their approach.
--The results are fairly compelling, and the benchmark indeed succeeds in yielding performance drops even from advanced models.

**Weaknesses:**

--Experimental results could have been deeper in the main text. It is for this reason that I am not inclined to give the paper a stellar rating.
--The approach is simple and has some nice properties, but I am not too sure about its sensitivity and robustness. I felt inadequate attention was paid to this in the paper.

**Questions:**

None.

---

> ### Author Response · Authors · 2024-11-15
> **Response to Review**
>
> Thank you for reviewing our paper. We are glad that you found our paper impressive and the results compelling. Please find our response to specific comments below.
>
> **[W1]** *Experimental results could have been deeper…but I am not too sure about its sensitivity and robustness*
>
> We have now increased the size of the code and math datasets by more than twice as much. Further, we have shown that our framework is robust enough to be applied to three very different types of tasks, and succeed in generating challenging problems for all of them. We have also shown that we can generate data using powerful models (such as GPT-4o) and comparatively weaker models (such as Llama-3.1-8B). We hope our experiments with increased data size address your concern about robustness. If you would like more analysis, could you please specify the kind of analysis you would like?
>
> We were encouraged to see that you had given a score of 8, and indeed your review (“It is for this reason that I am not inclined to give the paper a stellar rating”) reflects that. We kindly ask you to consider raising the score if your concerns are addressed.

---

> > ### Author Response · Authors · 2024-11-21
> > **Request to engage in discussion**
> >
> > We have now significantly increased the size of the benchmarks. Further, we have experimented with another prompt type for CHASE-Math, provided a more fine-grained evaluation for CHASE-QA, and compared the difficulties with other challenging datasets in the corresponding domains (see Appendix C).
> >
> > We kindly ask you to consider raising your score if your concerns are addressed or to kindly engage in a discussion with us.

---

> ### Author Response · Authors · 2024-11-25
> **Request to engage in discussion**
>
> We kindly ask you to increase your score if your concerns are addressed or to engage in a discussion with us. There are only 2 days left in the discussion.

---

> ### Author Response · Authors · 2024-11-29
> **Request to engage in discussion**
>
> Dear Reviewer,
>
> We request you to kindly read our response and adjust your score if your concerns are addressed.

---

### Author Response · Authors · 2024-11-24
**Request to read author responses and adjust scores**

Dear reviewers,

We have provided detailed responses to address your concerns and questions. Further, we have carried out many new experiments to support our claims and arguments. Since there are only 3 more days of discussion left, we request you to kindly look at our responses. If your concerns are addressed, we kindly ask you to increase your score. If not, we encourage you to respond to us and engage in a discussion.

---

### Author Response · Authors · 2024-12-04
**Summary of Discussion and Final Comments**

**Reviewer CAvC**

The reviewer did not participate in the discussion.
In their review, they made a vague comment about “not being sure about sensitivity and robustness”. This statement is not backed by any concrete point or weakness and we have responded by explaining how our method and experiments are actually quite robust (we showed it works for multiple domains, it is scalable, works with weaker LLMs, etc).
Further, we highlight that **the reviewer had initially given a score of 8, and then lowered it to 6 on the day the reviews were released without changing the review text and without waiting for our response**. This behaviour is quite unethical and we request the AC to consider their previous score in their deliberation.

**Reviewer gKcy**

We believe the reviewer has some unfounded concerns which we have sufficiently addressed in our responses.

**Size of data.** As already explained in detail in our responses, for long-context benchmarks, ~500 data points is the standard size. Asking for more data than that is against the spirit of open and accessible science since it will be prohibitively expensive to run models on the benchmark. Moreover, we have provided evidence of popular benchmarks and past papers published at ICLR whose sole focus is proposing a benchmark of just ~200-700 examples. Lastly, the benchmarks are not the main contribution of this paper, rather it is the underlying approach that was able to automatically generate the data. Hence, we strongly believe this concern to be unreasonable.

**Correctness.** We have already clarified that the generated data for code and math domains is completely correct. There is a possibility of a small percentage (~6%) of errors in the QA benchmark. We address this by providing softer metrics of evaluation (see detailed response) and by noting the large gaps between performances of models. Further, we highlight that the level of correctness of generated data achieved by our method is far superior to other baselines.

**Comparison of difficulty.** We have provided comparison of performance of LLMs on our benchmarks against other difficult benchmarks in the respective domains. The reviewer has failed to specify what other kind of “difficulty analysis” they were hoping to see. We believe it is clear from our results that our generated benchmarks are indeed quite challenging.

The reviewer had misunderstood some aspects of our paper such as the context size and fine-tuning experiments, which we have clarified in detail.

**Reviewer yxKN**

The reviewer has raised two concerns which we believe are not valid. We have already summarized our arguments for the size of data. The other point raised by this reviewer is that our experiments pertaining to the code domain do not reflect realistic scenarios. We believe this point is completely baseless and we have provided a detailed explanation in our response. Further note that the reviewer has no other standing concerns against our method or our experiments with the other 2 domains.

Overall, we feel that **reviewer gKcy and yxKN’s scores are quite disproportionate corresponding to their standing concerns**. They have provided no concrete counter-arguments to our responses. This work provides the first comprehensive approach to generate challenging, high-quality synthetic data for evaluation across multiple domains. The **reviewers agree that the problem that we study is critical** (CAvC, gKcy), the **paper is well-written** (CAvC, gKcy, yxKN), the **approach is novel** (gKcy), the **experiments are comprehensive** (yxKN), and the **results are compelling** (CAvC). None of the reviewers have pointed out any technical weaknesses in our proposed method or the main results (Table 1), which are the main contributions of this work.

---

### Meta-Review · Area_Chair_WPXM · 2024-12-23

**Metareview:**

This paper presents a systematic approach to synthesize challenging compositional problems for LLM evaluation in math, coding and general question answering. The core idea is to take a “bottom-up” approach to gradually compose simpler sub-tasks that are easier to verify to form more challenging benchmark problems. The authors showed that state-of-the-art LLMs only attain 40%-60% accuracy on the generated benchmarks, and claimed that such evaluation results demonstrated the effectiveness of the proposed approach in generating hard evaluation problems.

**Strengths:**

* The paper is generally well written (CAvC, yxKN) and well-structured with comprehensive appendices (gKcy). The method also has a clear motivation (CAvC, yxKN). “The problem addressed by this paper is critical to the evaluation of LLMs” (gKcy).

* The paper presents a “novel paradigm for data construction” (gKcy). The authors demonstrated the applicability of the benchmark synthesis approach on three distinct domains (CAvC).

* Comprehensive evaluation results “covering representative proprietary and open-source models” (yxKN). Reviewer CAvC also found the results “fairly compelling” as the benchmark “indeed succeeds in yielding performance drops even from advanced models”.

**Weaknesses:**

After the rebuttal period, there are several issues that are yet to be addressed.

First, while the results suggest that LLMs do not perform well on this dataset, the paper lacks intrinsic evaluation on the complexity and difficulty of the synthesized problems (gKcy). The authors could consider using well-established domain-specific metrics to measure problem complexity, such as using the number of lines or the size of ASTs to approximate program complexity for CHASE-CODE. As a general suggestion, while the reviewer did not specify the exact metrics to use in their review, the authors would have been more proactive in the response period and report any potentially reasonable metrics in order to better address the reviewer’s concern.

Another potential issue is the quality of the synthesized problems (gKcy). While we appreciate the authors’ effort in reviewing O(30 - 100) problems in the three datasets, given the total size of each dataset (around 500), it would be more convincing to carefully review at least 20% examples (i.e., 100 problems) in order to reach any statistically significant conclusions. The authors already demonstrated that the CHASE-MATH dataset is in relatively decent quality by reviewing 100 examples, and I strongly suggest the authors review a similar number of tasks in the other two datasets.

Next, as flagged by Reviewer yxKN, since there already exist high-quality, challenging datasets for repository-level code editing (SWE-bench), it is less clear that from an empirical perspective, what additional values would the new synthetic CHASE-Code dataset bring to the code LLM community. While we acknowledge that CHASE-Code focuses more on *generation* Instead of code editing, there are existing high-quality repo-level code generation benchmarks derived from real repository-level context, such as DevEval (https://arxiv.org/abs/2405.19856). I totally agree with the authors that the value of this paper lies more in the proposed data synthesis approach, instead of the datasets produced. However, a significant portion of this paper also comes from the value of the benchmarks the proposed approach synthesized, it is hard for me to assess the practical value and implications of CHASE-Code. In the future, maybe the authors could take the reviewer’s suggestion, and explore additional use cases or domains in coding where the proposed method could synthesize benchmarks with more practical value to the practitioners. On the other hand, the authors could also consider exploring whether your synthetic datasets like CHASE-Code could complement or correlate with existing benchmarks like SWE-Bench or LMSYS coding, or other datasets that require more laborious human efforts. In this way, the authors could more clearly demonstrate the value of CHASE as it provides a significantly more cost-effective approach to create novel code benchmarks.

—

Finally, I wanted to note that the authors’ attitude during the rebuttal period is unprofessional and might potentially violate ICLR code of ethics (“Researchers must show respect for colleagues, research participants …”). It is totally understandable that you may find reviewers might take a different perspective when judging your work, and it is critical to professionally resolve any concerns or misunderstandings via peaceful communication in a respectful manner. While I did not take this into consideration when rating your work this time, I wish the authors could bear this in mind in the future.

**Additional Comments On Reviewer Discussion:**

There are other issues, such as questions around the size of the datasets, which are addressed during the rebuttal period.

---

### Decision · Program_Chairs · 2025-01-22

Reject